# ProtoPairNet: Interpretable Regression through Prototypical Pair Reasoning

**Rose Gurung**
University of Maine
rose.gurung@maine.edu

**Ronilo Ragodos**
University of New Hampshire
Ronilo.Ragodos@unh.edu

**Chiyu Ma**
Dartmouth College
chiyu.ma.gr@dartmouth.edu

**Tong Wang**
Yale University
tong.wang.tw687@yale.edu

**Chaofan Chen**
University of Maine
chaofan.chen@maine.edu

## Abstract

We present Prototypical Pair Network (ProtoPairNet), a novel interpretable architecture that combines deep learning with case-based reasoning to predict continuous targets. While prototype-based models have primarily addressed image classification with discrete outputs, extending these methods to continuous targets, such as regression, poses significant challenges. Existing architectures which rely heavily on one-to-one comparison with prototypes lack the directional information necessary for continuous predictions. Our method redefines the role of prototypes in such tasks by incorporating prototypical pairs into the reasoning process. Predictions are derived based on the input's relative dissimilarities to these pairs, leveraging an intuitive geometric interpretation. Our method further reduces the complexity of the reasoning process by relying on the single most relevant pair of prototypes, rather than all prototypes in the model as was done in prior works. Our model is versatile enough to be used in both vision-based regression and continuous control in reinforcement learning. Our experiments demonstrate that ProtoPairNet achieves performance on par with its black-box counterparts across these tasks. Comprehensive analyses confirm the meaningfulness of prototypical pairs and the faithfulness of our model's interpretations, and extensive user studies highlight our model's improved interpretability over existing methods.

## 1 Introduction

Deep neural networks have achieved remarkable success across various domains, from advancing autonomous vehicles [31] to supporting critical decisions in healthcare [5, 46, 39] and criminal justice [6]. As these applications expand into increasingly sensitive and high-stakes areas, the demand for models with transparent and interpretable decision-making processes has become crucial. Prototype-based models [11, 44, 33, 26, 12, 43, 28, 23], which are inherently interpretable by design, address this challenge through **case-based reasoning** [1], a natural form of human reasoning. These models generally perform **one-to-one** comparisons between an input image and prototypical cases learned during training. They then aggregate evidence of similarity to these prototypes, using a linear layer to arrive at the final classification decision.

Despite recent advances in prototype-based models for image classifications with discrete targets, extending these models to tasks with continuous targets, such as regression [24, 16, 21] face two primary issues: First, one-to-one comparisons between an input image and individual prototypes only tell us that the input image is similar to some prototypes and thus should have a label similar to the labels of those prototypes. This is insufficient for regression, where

predictions in continuous space would require additional context, such as whether the output should be greater or less than each prototype's label. This is shown at the top of Figure 1, where the input image has comparable similarities to both prototypes, making it unclear whether its label should be greater or smaller than those of the individual prototypes.

Second, using a linear layer at the end to combine prototype labels via a similarity-weighted average can cause unwarranted deviations. For example, even when an input matches a prototype exactly, the predicted value may deviate from the prototype's label due to interference from irrelevant prototypes, even if their contributions to the output is small. An illustration for the second issue can be found in Appendix B.

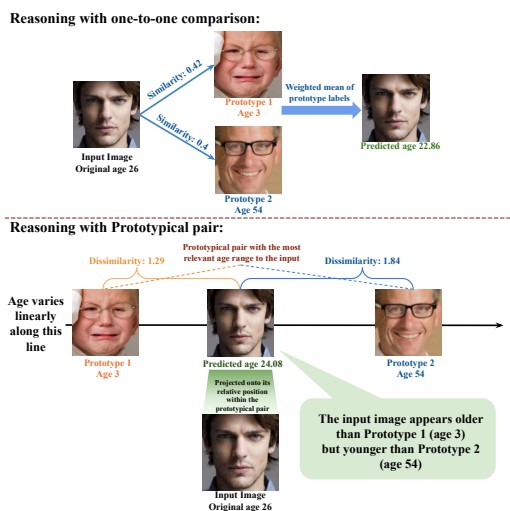

Inspired by these challenges, we propose the Prototypical Pair Network (ProtoPairNet), a novel architecture that redefines the role of prototypes in regression tasks through case-based reasoning. Unlike traditional one-to-one image-to-prototype comparisons, ProtoPairNet introduces **prototypical pairs** – pairs of learned prototypes that jointly govern the direction and magnitude of the predicted label. Each prototypical pair defines an axis along which the regression output varies linearly. By comparing an input image with the most relevant prototype pairs identified by the model, ProtoPairNet captures both proximity to the axis defined by the prototype pair and relative positioning within the range defined by the pair. As illustrated in the bottom of Figure 1 for age prediction, the input image is projected onto a line defined by a pair of prototypes in the latent space. Our model is

Figure 1: Reasoning process of how prototype-based models predict the age of an input image: traditional one-to-one comparison (top) vs. prototypical pairs reasoning (bottom). One-to-one comparisons lack directional cues on whether the predicted label should exceed or fall below the prototype's label, leading to ambiguity in explanations.

designed to ensure that age varies linearly along this line and therefore it can use the input's relative position between the prototypes to determine the predicted age. This intuitive geometric interpretation of prototypes further eliminates the need for a separate linear layer, allowing predictions to rely solely on relevant prototypical pairs. This refined representation enables a more insightful and effective reasoning process in continuous spaces, making ProtoPairNet particularly suited for regression tasks.

To demonstrate the versatility of ProtoPairNet, we evaluate it on two distinct domains: age prediction (a supervised learning task) and car racing (a behavioral cloning task with a reinforcement learning expert). Experimental results show that ProtoPairNet achieves *performance competitive with black-box baselines* in both settings. Meanwhile, multiple user studies further highlight that *ProtoPairNet improves interpretability* over conventional prototype-based models relying on one-to-one prototype comparisons. Additionally, the illustrations of our reasoning processes and global analyses empirically demonstrate the *consistency and faithfulness* of the prototype representations.

**Related Work**

*Posthoc* explanation methods for computer vision, such as activation maximization [13, 27], image perturbation [14, 19, 29], attention heatmaps [2, 49, 37], and saliency visualizations [38, 4, 47, 40, 41], often fail to faithfully explain the reasoning processes of deep neural networks, as their explanations may not accurately reflect the model's decision-making [3, 32].

Prototype-based approaches offer a transparent prediction process through **case-based reasoning**. These models compare a small set of learned latent feature representations called prototypes (the "cases") with the latent representations of a test image to perform classification. For example, the Prototypical Part Network (ProtoPNet) [11] employs **one-to-one comparisons** with class-specific prototypes using $L_2$ distance. Each prototype is trained to closely resemble feature patches from its own class while being dissimilar to patches from other classes. The Transparent Embedding Space Network (TesNet) [44] modifies the original ProtoPNet by utilizing a cosine similarity metric to

compute similarities between image patches and prototypes in a latent space. Deformable ProtoPNet [12] decomposes the prototypes into smaller patches to capture pose variations. Other works [33, 26, 34] reduce the reliance on class-specific prototypes, thereby minimizing the number of prototypes required. Recent advancements have focused on improving quality of prototypes [7, 45, 42, 9], enhancing interpretability through modified architectures [22, 28], learning prototypes as distributions [35, 10], and extending prototype-based models to vision transformers [23], text classification [25, 18], and reinforcement learning with discrete action spaces [30].

However, existing work on prototype-based models has predominantly focused on predicting discrete outcomes, and very few address the prediction of continuous outcomes such as regression. Among prototype-based models addressing regression tasks, Hyperspherical Prototype Networks [24] sacrifice interpretability by defining prototypes as abstract points disconnected from real data, thereby reducing the transparency of predictions. ExPeRT [17] and INSightR-Net [16] mitigates this by anchoring prototypes to real data and calculating predictions as similarity-weighted averages of all prototype labels, akin to ProtoPNet [11] and its variants. However, this approach in regression tasks remains vulnerable to interference from unrelated prototypes, potentially leading to inaccurate predictions. Prototype-Wrapper Networks (PWNet) [21], which can be used for continuous control in reinforcement learning, face limitations in adaptability and scalability due to their reliance on manual prototype selection and fixed layer weights. Furthermore, they inherit the same vulnerability from similarity-weighted averaging, where an input identical to a prototype can yield a different prediction due to the influence of other prototypes (see Appendix B). These works primarily rely on one-to-one comparisons, which only provides limited information for regression tasks and introduces ambiguity into the reasoning process. In contrast, our model makes regression predictions by comparing input images with the most relevant prototypical pairs it has learned. **Our ProtoPairNet enhances interpretability by reducing dependence on similarity-weighted averages of all prototype labels.**

## 2 Method

### 2.1 Architecture Overview

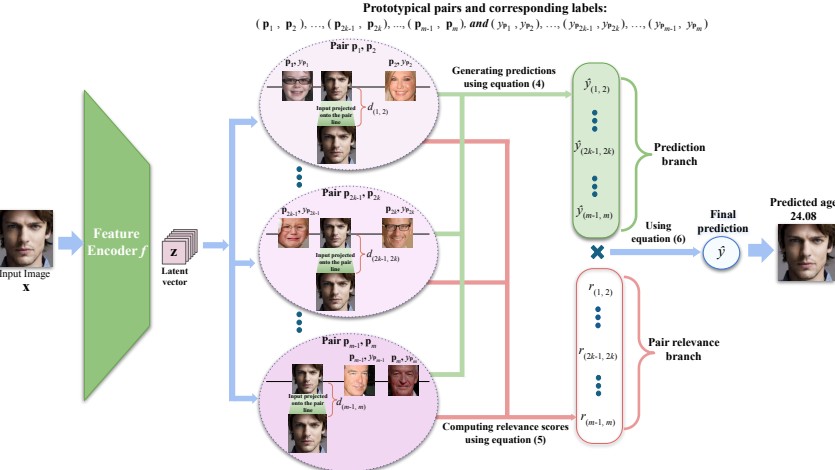

Figure 2: Model architecture for ProtoPairNet. The pink bubbles show the comparison of each prototypical pair with the input in the latent space. This information is used by both the prediction branch (green) and the pair relevance branch (red) to produce the final prediction.

Figure 2 provides an overview of the ProtoPairNet architecture. Our model consists of a feature encoder $f$ that encodes an input image $\mathbf{x}$ to a latent vector $\mathbf{z} = f(\mathbf{x})$, followed by two branches: a prediction branch and a pair relevance branch. Both branches utilize prototypical pairs $(\mathbf{p}_1, \mathbf{p}_2), ..., (\mathbf{p}_{2k-1}, \mathbf{p}_{2k}), ..., (\mathbf{p}_{m-1}, \mathbf{p}_m)$, where $m$ is the number of prototypes, and is an even integer, resulting in $m/2$ prototypical pairs. For each $k = 1, ..., m/2$, the $k$-th prototypical pair $(\mathbf{p}_{2k-1}, \mathbf{p}_{2k})$ defines an axis, which we call the $k$-**th ProtoPair axis**, along which the predicted label increases linearly in the direction of $\mathbf{p}_{2k} - \mathbf{p}_{2k-1}$. Each prototypical pair $(\mathbf{p}_{2k-1}, \mathbf{p}_{2k})$ is also associated with the prototypical labels $(y_{\mathbf{p}_{2k-1}}, y_{\mathbf{p}_{2k}}) \in \mathbb{R}^2$, which denote the ground-truth labels of

the corresponding prototypes. Both the prototypical pairs and their associated labels are learned from the training data. Since the focus of this work is to introduce a new paradigm of reasoning using prototypical pairs for continuous target predictions, we have opted to use prototypes that have the same dimensions as the latent vector $\mathbf{z}$, which represents the entire input rather than a localized part. The extension to prototypical parts will be investigated in future work.

Given an input image $\mathbf{x}$, our ProtoPairNet compares its latent representation $\mathbf{z} = f(\mathbf{x})$ with $m/2$ prototypical pairs. For each of the $k$-th prototypical pairs $(\mathbf{p}_{2k-1}, \mathbf{p}_{2k})$, the latent vector $\mathbf{z}$ is first projected onto the $k$-th ProtoPair axis. Let $\mathbf{z}'_{(2k-1,2k)}$ denote the projected point (see Figure 3). Since our model is designed to ensure that the target labels vary linearly along the ProtoPair axis, the predicted label of $\mathbf{x}$ depends on the position of $\mathbf{z}'_{(2k-1,2k)}$ along the $k$-th ProtoPair axis. We then use the relative distances of $\mathbf{z}'_{(2k-1,2k)}$ with $\mathbf{p}_{2k-1}$ and $\mathbf{p}_{2k}$ to predict a label $\hat{y}_{(2k-1,2k)}$. The prediction branch of our model makes $m/2$ predictions $\hat{y}_{(1,2)}, ..., \hat{y}_{(m-1,m)}$, one for each prototypical pair. Section 2.2 provides more details on the prediction branch.

The pair relevance branch of our model identifies the most relevant prototypical pair for a prediction, by first calculating the squared $L_2$ distance between the latent vector $\mathbf{z}$ and each $k$-th ProtoPair axis. This is given by $d_{(2k-1,2k)} = \|\mathbf{z} - \mathbf{z}'_{(2k-1,2k)}\|_2^2$. A one-hot relevance score vector $\mathbf{r}$ is constructed by applying Gumbel-softmax to $-d_{(1,2)}, ..., -d_{(m-1,m)}$ to generate the pair relevance score $r_{(1,2)}, ..., r_{(m-1,m)}$. Since Gumbel-softmax approximates the $\mathrm{argmax}$, our model assigns a relevance of $1$ to the prototypical pair that is closest to the latent vector $\mathbf{z}$. The final prediction $\hat{y}$ corresponds to the output of the prediction branch associated with the closest prototypical pair. Section 2.3 describes the pair relevance branch in more detail.

## 2.2 Prediction Branch

As shown in Figure 3, we consider a pair of prototypes $(\mathbf{p}_j, \mathbf{p}_{j'})$, with $j = 1, 3, \ldots, m-1$ and $j' = j + 1$. For the $k$-th prototypical pair, we have $j = 2k - 1$ and $j' = 2k$. Each prototype $\mathbf{p}_j$ is associated with a prototypical label $y_{\mathbf{p}_j} \in \mathbb{R}$. The geometric intuition is that, for each latent vector $\mathbf{z}$, and each pair of prototypes $(\mathbf{p}_j, \mathbf{p}_{j'})$ with corresponding labels $(y_{\mathbf{p}_j}, y_{\mathbf{p}_{j'}})$, we predict an output $\hat{y}_{(j,j')}$ by first projecting $\mathbf{z}$ onto the $k$-th ProtoPair axis along the vector $\mathbf{p}_{j'} - \mathbf{p}_j$, as $\mathbf{z}'_{(j,j')}$. Mathematically, this projection is given by:

$$\mathbf{z}'_{(j,j')} = \mathbf{p}_j + \|\mathbf{z} - \mathbf{p}_j\|_2 \cos\theta \frac{\mathbf{p}_{j'} - \mathbf{p}_j}{\|\mathbf{p}_j - \mathbf{p}_{j'}\|_2}, \quad (1)$$

where $\theta$ denotes the angle between the vector $\mathbf{z} - \mathbf{p}_j$ and the vector $\mathbf{p}_{j'} - \mathbf{p}_j$.

Figure 3: Geometric interpretation of a prototype pair and a latent vector.

Two key assumptions underlie this framework: (1) the labels vary linearly along the $k$-th ProtoPair axis, and (2) latent representations for the same label lie on a shared hyperplane. As a result, $\mathbf{z}$ and $\mathbf{z}'_{(j,j')}$ will have the same predicted label $\hat{y}_{(j,j')}$. Using these assumptions and equation (1), the prediction $\hat{y}_{(j,j')}$ is estimated as:

$$\hat{y}_{(j,j')} = y_{\mathbf{p}_j} + \|\mathbf{z} - \mathbf{p}_j\|_2 \cos\theta \frac{y_{\mathbf{p}_{j'}} - y_{\mathbf{p}_j}}{\|\mathbf{p}_j - \mathbf{p}_{j'}\|_2}. \quad (2)$$

Since we have

$$\|\mathbf{z} - \mathbf{p}_j\|_2 \cos\theta = \|\mathbf{z} - \mathbf{p}_j\|_2 \frac{(\mathbf{z} - \mathbf{p}_j)^T}{\|\mathbf{z} - \mathbf{p}_j\|_2} \frac{(\mathbf{p}_{j'} - \mathbf{p}_j)}{\|\mathbf{p}_j - \mathbf{p}_{j'}\|_2} = \frac{[(\mathbf{z} - \mathbf{p}_j)^T (\mathbf{p}_{j'} - \mathbf{p}_j)]}{\|\mathbf{p}_j - \mathbf{p}_{j'}\|_2}, \quad (3)$$

we can substitute the right hand side of equation (3) into the equation (2) for $\hat{y}_{(j,j')}$, obtaining:

$$\hat{y}_{(j,j')} = y_{\mathbf{p}_j} + \frac{[(\mathbf{z} - \mathbf{p}_j)^T (\mathbf{p}_{j'} - \mathbf{p}_j)]}{\|\mathbf{p}_j - \mathbf{p}_{j'}\|_2} \frac{y_{\mathbf{p}_{j'}} - y_{\mathbf{p}_j}}{\|\mathbf{p}_j - \mathbf{p}_{j'}\|_2} = y_{\mathbf{p}_j} + \frac{[(\mathbf{z} - \mathbf{p}_j)^T (\mathbf{p}_{j'} - \mathbf{p}_j)] (y_{\mathbf{p}_{j'}} - y_{\mathbf{p}_j})}{(\mathbf{p}_{j'} - \mathbf{p}_j)^T (\mathbf{p}_{j'} - \mathbf{p}_j)}. \quad (4)$$

Using equation (4), the prediction branch takes a latent representation $\mathbf{z}$ as input and outputs, for each prototypical pair $(\mathbf{p}_{2k-1}, \mathbf{p}_{2k})$, a continuous label $\hat{y}_{(2k-1,2k)}$.

This geometric formulation resolves both of the challenges (discussed in Section 1) of applying existing prototype-based models to regression. By using equation (4) to compute predictions, our model ensures that the predicted label varies linearly along the ProtoPair axis defined by $(\mathbf{p}_j, \mathbf{p}_{j'})$. In particular, equation (4) indicates whether the prediction should be greater or less than a given prototypical label, based on the input's relative position along the axis in the latent space. Additionally, this formulation ensures that when an input matches a prototype exactly, the prediction will match the prototype's label – mathematically, when we have $\mathbf{z} = \mathbf{p}_j$ or $\mathbf{z} = \mathbf{p}_{j'}$, equation (4) yields $\hat{y}_{(j,j')} = y_{\mathbf{p}_j}$ or $\hat{y}_{(j,j')} = y_{\mathbf{p}_{j'}}$, respectively. In the next section, we will discuss how we can scale our model to handle more complex tasks requiring multiple prototypical pairs.

## 2.3 Pair Relevance Branch

Since ProtoPairNet can have multiple prototypical pairs, each producing a different prediction label, it is necessary to identify a pair that is most suited to making the final prediction. This motivates us to define the pair relevance branch, which estimates a relevance score for each prototypical pair. We further design our model to consider the distance between a latent vector $\mathbf{z}$ and each ProtoPair axis, such that the closest ProtoPair axis identifies the most relevant pair for making the final prediction. Mathematically, given a latent vector $\mathbf{z}$, the pair relevance branch first computes its squared $L_2$ distance to each $k$-th ProtoPair axis: $d_{(2k-1,2k)} = \left\| \mathbf{z} - \mathbf{z}'_{(2k-1,2k)} \right\|_2^2$, for all $k = 1, ..., m/2$. Then, it computes a relevance vector $\mathbf{r} = [r_{(1,2)}, ..., r_{(m-1,m)}] = \text{Gumbel-softmax}([-d_{(1,2)}, ..., -d_{(m-1,m)}])$. We use Gumbel-softmax [20] as a differentiable argmax "approximation": it has a temperature parameter $\tau$ with the property that as $\tau \to 0$, the output of Gumbel-softmax converges to one-hot vectors. Our use of Gumbel-softmax to compute $r_{(2k-1,2k)}$ for the $k$-th prototypical pair is given by

$$r_{(2k-1,2k)} = \frac{\exp\big[(-d_{(2k-1,2k)} + g_{(2k-1,2k)})/\tau\big]}{\sum_{k'=1}^{m/2} \exp\big[(-d_{(2k'-1,2k')} + g_{(2k'-1,2k')})/\tau\big]}, \tag{5}$$

where $g_{(2k'-1,2k')}$ are i.i.d. random variables sampled from the Gumbel distribution [15] and $\tau$ is the softmax temperature. The final prediction of ProtoPairNet is given by

$$\hat{y} = \sum_{k=1}^{m/2} r_{(2k-1,2k)} \, \hat{y}_{(2k-1,2k)}. \tag{6}$$

The use of Gumbel-softmax means that the final prediction $\hat{y}$ is essentially the prediction made by a single, most relevant prototype pair. This not only ensures decision sparsity, but also ensures that the final prediction does not suffer from interference of other (irrelevant) prototype pairs.

## 2.4 Training Algorithm

The training of ProtoPairNet is divided into three stages: (1) optimization via stochastic gradient descent (SGD), (2) projection of prototypical pairs onto training instances, and (3) post-projection optimization. Stage 3 is crucial for ensuring alignment between the projected prototypes and the rest of the network, as even slight misalignment in the latent space can lead to significant errors when predicting continuous values. The feature encoder $f$ used in these stages is pretrained on the task-specific dataset beforehand. We use $\mathcal{D} = \{(\mathbf{x}^{(i)}, y^{(i)})\}_{i=1}^n$ to denote a training dataset.

**Stage 1. Optimization via stochastic gradient descent (SGD):** In the first training stage, we aim to learn a meaningful latent space where the latent representations of data points are clustered around semantically similar prototypes and each of them is close to at least one ProtoPair axis. We initialize the prototypical labels using $k$-means clustering over the labels of the training dataset, to ensure that they represent distinct and well-distributed regions. In this stage, the prototypical labels $y_{\mathbf{p}_1}, ..., y_{\mathbf{p}_m}$ are fixed, and the parameters in the encoder $f$ and the prototypical pairs $(\mathbf{p}_1, \mathbf{p}_2), ..., (\mathbf{p}_{m-1}, \mathbf{p}_m)$ are optimized, by minimizing a loss function consisting of a mean absolute error (MAE) or mean square error (MSE), a cluster loss, an axis distance loss, and a diversity loss.

We use a cluster loss adapted for regression defined as:

$$\mathcal{L}_{\text{Clst}} = \frac{1}{n} \sum_{i=1}^n \sum_{j=1}^m \exp\left( -\eta(y^{(i)} - y_{\mathbf{p}_j})^2 \right) \left\| \mathbf{z}^{(i)} - \mathbf{p}_j \right\|_2^2,$$

where $\mathbf{z}^{(i)} = f(\mathbf{x}^{(i)})$ is the latent representation obtained from the encoder $f$ and $y^{(i)}$ is the label of the input image $\mathbf{x}^{(i)}$. Also, $\eta > 0$ is a predefined scaling factor, and all other notations are defined in previous sections. The cluster loss encourages the latent representation $\mathbf{z}^{(i)}$ of a training image $\mathbf{x}^{(i)}$ to be close to prototypes of similar labels, as the squared distance between $\mathbf{z}^{(i)}$ and each prototype $\mathbf{p}_j$ is weighted by the closeness of the target label $y^{(i)}$ to the prototypical label $y_{\mathbf{p}_j}$. This weight is larger when the target label $y^{(i)}$ is closer to the prototypical label $y_{\mathbf{p}_j}$, decaying exponentially as the target label deviates from the prototypical label.

To enforce the geometric structure required for our approach, we use an axis distance loss defined as:

$$\mathcal{L}_{\text{AxisDist}} = \frac{1}{n} \sum_{i=1}^{n} \left\| \mathbf{z}^{(i)} - \mathbf{z}'^{(i)}_{(2k^*-1, 2k^*)} \right\|_2^2,$$

with $k^* = \arg\min_k \min \left( |y^{(i)} - y_{\mathbf{p}_{2k-1}}|, |y^{(i)} - y_{\mathbf{p}_{2k}}| \right)$, and $\mathbf{z}'^{(i)}_{(2k^*-1, 2k^*)}$ being the projection of $\mathbf{z}^{(i)}$ onto the $k^*$-th ProtoPair axis. The axis distance loss encourages the latent representation $\mathbf{z}^{(i)}$ to be close to the ProtoPair axis defined by the $k^*$-th prototypical pair. This pair is chosen as the one whose associated prototypical label range is the closest to the target label $y^{(i)}$. While the cluster loss ensures that the latent representation of a training image stays close to individual prototypes, the axis distance loss ensures that the latent representation of a training image stays close to a ProtoPair axis, thereby facilitating the identification of the most relevant pair of prototypes.

Additionally, we use an intra-pair diversity loss that encourages prototypes within each pair to be distinct: $\mathcal{L}_{\text{Div}} = -\sum_{k=1}^{m/2} \|\mathbf{p}_{2k-1} - \mathbf{p}_{2k}\|_2^2$. Overall, this training stage aims to minimize the total loss:

$$\mathcal{L}_{\text{total}} = \mathcal{L}_{\text{MAE/MSE}} + \lambda_1 \mathcal{L}_{\text{Clst}} + \lambda_2 \mathcal{L}_{\text{AxisDist}} + \lambda_3 \mathcal{L}_{\text{Div}}. \tag{7}$$

We use MAE for Age Prediction, MSE for Car Racing, and $\lambda_1, \lambda_2, \lambda_3$ are hyperparameters.

**Stage 2. Projection of prototypical pairs onto training instances:** To visualize the prototypical pairs $\{(\mathbf{p}_{2k-1}, \mathbf{p}_{2k})\}_{k=1}^{m/2}$, we need to project them onto pairs of latent representations of training instances. However, simply replacing prototypes with their nearest latent representations as was done in ProtoPNet [11] is undesirable, because this could drastically change the direction of the ProtoPair axes, leading to incorrect predictions and performance degradations. To address this issue, we propose a novel prototype projection algorithm, which evaluates whether the latent representations of a pair of training images $(\mathbf{x}^{(i)}, \mathbf{x}^{(i')})$ can serve as a good candidate for projecting the $k$-th prototype pair $(\mathbf{p}_{2k-1}, \mathbf{p}_{2k})$ using three metrics:

(i) the cosine similarity $\alpha_k^{(i,i')}$ between the vectors $\mathbf{p}_{2k-1} - \mathbf{p}_{2k}$ and $\mathbf{z}^{(i)} - \mathbf{z}^{(i')}$: $\alpha_k^{(i,i')} = \frac{(\mathbf{p}_{2k-1} - \mathbf{p}_{2k})^T (\mathbf{z}^{(i)} - \mathbf{z}^{(i')})}{\|\mathbf{p}_{2k-1} - \mathbf{p}_{2k}\|_2 \|\mathbf{z}^{(i)} - \mathbf{z}^{(i')}\|_2}$, where $\mathbf{z}^{(i)} = f(\mathbf{x}^{(i)})$ and $\mathbf{z}^{(i')} = f(\mathbf{x}^{(i')})$ denote the latent representations of $\mathbf{x}^{(i)}$ and $\mathbf{x}^{(i')}$, respectively;

(ii) a diversity score $\beta^{(i,i')}$ which measures the differences between the latent representations $\mathbf{z}^{(i)}$ and $\mathbf{z}^{(i')}$, as well as the labels $y^{(i)}$ and $y^{(i')}$ of the training image pair $(\mathbf{x}^{(i)}, \mathbf{x}^{(i')})$: $\beta^{(i,i')} = |y^{(i)} - y^{(i')}| \|\mathbf{z}^{(i)} - \mathbf{z}^{(i')}\|_2^2$;

(iii) the average axis distance $\gamma_k^{(i,i')}$ which measures the average of the squared distances between the $k$-th ProtoPair axis and the latent representations $\mathbf{z}^{(i)}$ and $\mathbf{z}^{(i')}$: $\gamma_k^{(i,i')} = \frac{1}{2}(\|\mathbf{z}^{(i)} - \mathbf{z}'^{(i)}_{(2k-1,2k)}\|_2^2 + \|\mathbf{z}^{(i')} - \mathbf{z}'^{(i')}_{(2k-1,2k)}\|_2^2)$, where $\mathbf{z}'^{(i)}_{(2k-1,2k)}$ and $\mathbf{z}'^{(i')}_{(2k-1,2k)}$ are the projections of $\mathbf{z}^{(i)}$ and $\mathbf{z}^{(i')}$ onto the $k$-th ProtoPair axis, respectively.

The cosine similarity $\alpha_k^{(i,i')}$ measures whether replacing the prototype pairs $(\mathbf{p}_{2k-1}, \mathbf{p}_{2k})$ with $(\mathbf{z}^{(i)}, \mathbf{z}^{(i')})$ changes the direction of the $k$-th ProtoPair axis, with higher values indicating less changes in the direction of the ProtoPair axis. The diversity score $\beta^{(i,i')}$ is needed because the prediction becomes numerically unstable if $\mathbf{z}^{(i)}$ and $\mathbf{z}^{(i')}$ are too close to each other, and becomes difficult to interpret if $y^{(i)}$ is too close to $y^{(i')}$. The average axis distance $\gamma_k^{(i,i')}$ is used to find training image pairs that are close to the $k$-th ProtoPair axis. For each prototype pair $(\mathbf{p}_{2k-1}, \mathbf{p}_{2k})$, our projection algorithm selects a training image pair $(\mathbf{x}^{(i)}, \mathbf{x}^{(i')})$ that maximizes the cosine similarity $\alpha_k^{(i,i')}$, while

Table 1: Age prediction results. Comparision of ProtoPairNet with baseline black-box models (without prototypes) in terms of MAE and $R^2$ scores across different architectures.

| Architecture | ResNet50 | | EfficientNet-B0 | | VGG19 | | ViT-Small (Patch 16) | |
|---|---|---|---|---|---|---|---|---|
| | MAE | $R^2$ Score | MAE | $R^2$ Score | MAE | $R^2$ Score | MAE | $R^2$ Score |
| Baseline | $4.71 \pm 0.09$ | $0.86 \pm 0.004$ | $4.82 \pm 0.08$ | $0.86 \pm 0.004$ | $4.73 \pm 0.07$ | $0.86 \pm 0.005$ | $4.81 \pm 0.06$ | $0.86 \pm 0.004$ |
| **ProtoPairNet (Ours)** | $\mathbf{4.59 \pm 0.01}$ | $\mathbf{0.87 \pm 0.001}$ | $4.88 \pm 0.03$ | $0.86 \pm 0.0006$ | $\mathbf{4.57 \pm 0.01}$ | $\mathbf{0.87 \pm 0.001}$ | $\mathbf{4.63 \pm 0.04}$ | $\mathbf{0.87 \pm 0.001}$ |

ensuring that the diversity score $\beta^{(i,i')}$ is sufficiently large and the average axis distance $\gamma_k^{(i,i')}$ is sufficiently small. To reduce the computational complexity, we apply the prototype projection procedure to 50 randomly sampled training image pairs in each training batch.

Once a training image pair $(\mathbf{x}^{(i)}, \mathbf{x}^{(i')})$ is selected for the prototype pair $(\mathbf{p}_{2k-1}, \mathbf{p}_{2k})$, we update the prototype pair to $(\mathbf{z}^{(i)}, \mathbf{z}^{(i')})$ and update the corresponding prototypical labels $(y_{\mathbf{p}_{2k-1}}, y_{\mathbf{p}_{2k}})$ using the ground-truth labels $(y^{(i)}, y^{(i')})$ of the training image pair. We also store the original training images $\mathbf{x}^{(i)}$ and $\mathbf{x}^{(i')}$ as $\mathrm{Vis}(\mathbf{p}_{2k-1})$ and $\mathrm{Vis}(\mathbf{p}_{2k})$, where $\mathrm{Vis}$ denotes the pixel-space visualization of the corresponding prototype. More details of the projection algorithm, including the pseudo-code, can be found in Appendix G.

**Stage 3. Post-projection optimization:** In this stage, we fix the prototype visualizations $\mathrm{Vis}(\mathbf{p}_1)$, ..., $\mathrm{Vis}(\mathbf{p}_m)$ and minimize the same objective defined in equation (7) as in stage 1, but subject to the constraints $\mathbf{p}_j = f(\mathrm{Vis}(\mathbf{p}_j))$ for all $j = 1, ..., m$. This is implemented in code by storing $\mathrm{Vis}(\mathbf{p}_j)$ as parameters in the network with no gradients, and using only $f(\mathrm{Vis}(\mathbf{p}_j))$ in the model computations.

This stage is necessary because projecting prototypical pairs onto real data instances may introduce abrupt shifts in their latent space positions. Fine-tuning the model parameters, with the prototypical pairs fixed at the latent representations of their visualizations, ensures that these adjustments integrate smoothly with the overall latent structure and model parameters while preserving the semantic integrity of the projected prototypical pairs.

## 3 Experiments

### 3.1 Case Study 1: Age Prediction

In this case study, we evaluate ProtoPairNet on the UTKFace dataset [48] for age prediction, using only the age labels from 23,702 facial images. Ages above 84 are excluded due to the limited number of instances in the dataset for these older ages. We apply online augmentations—random rotation, flipping, and color jitter—only during training stages 1 and 3; prototype projection (stage 2) is performed on the original (unaugmented) dataset. The training set contains 18,615 images. We use 5 prototype pairs (10 prototypes total) in our main experiments, with ablation studies on loss components and prototype configurations presented in Appendix E.

We report the MAEs and $R^2$ scores achieved by ProtoPairNet and baseline models (without proto-types) across various architectures, averaged over three runs, in Table 1. Experiments were conducted using widely adopted convolutional neural network (CNN) architectures, including ResNet-50, EfficientNet-B0, and VGG-19, as well as the Vision Transformer (ViT-Small with a patch size of 16). All backbone models were initialized with pre-trained weights from the ImageNet dataset and then finetuned with the UTKFace dataset. Our results, summarized in Table 1, demonstrate that ProtoPairNet consistently delivers competitive performance across multiple backbone architectures. **For ResNet-50, VGG19, and ViT-Small (Patch 16) backbones, ProtoPairNet achieves lower MAEs and higher $R^2$ scores, demonstrating superior performance with these architectures while maintaining comparable performance for EfficientNet-B0.** Furthermore, ProtoPairNet introduces interpretability across all architectures, highlighting its ability to balance strong predictive performance with enhanced interpretability for regression tasks.

We also compare ProtoPairNet with other prototype-based regression models for age prediction. As shown in Table 2, **ProtoPairNet outperforms HPN** [24], **INSightR-Net** [16], and **ExPeRT** [17] **in both MAE and $R^2$ score**. HPN was evaluated using its regression variant with two label prototypes, while INSightR-Net and ExPeRT used 10 prototypes to match our 5 prototype pairs. All models use a ResNet-50 backbone and were finetuned using the same protocol as ProtoPairNet. Although HPN performs reasonably (MAE = 5.08), its prototypes are not visualizable and thus lack interpretability.

For INSightR-Net, we followed the official setup with backbone pretraining and $1 \times 1$ patch-based prototypes. To conduct a fair comparison with our ProtoPair-Net's non-patch-based design, we also trained an INSightR-Net using full latent representations as prototypes. For ExPeRT, we used their original configuration with full-image latent prototypes.

Table 2: Comparison of ProtoPairNet with other prototype-based ResNet50 models for regression using MAE and $R^2$ scores.

| Model | MAE | $R^2$ score |
|---|---|---|
| HPN | $5.08 \pm 0.03$ | $0.84 \pm 0.001$ |
| INSightR-Net ($1 \times 1$) | $14.01 \pm 0.01$ | $-0.03 \pm 0.001$ |
| INSightR-Net (full) | $14.81 \pm 0.40$ | $-0.08 \pm 0.07$ |
| ExPeRT | $17.36 \pm 2.79$ | $-0.37 \pm 0.41$ |
| ProtoPairNet (Ours) | $\mathbf{4.59 \pm 0.01}$ | $\mathbf{0.87 \pm 0.001}$ |

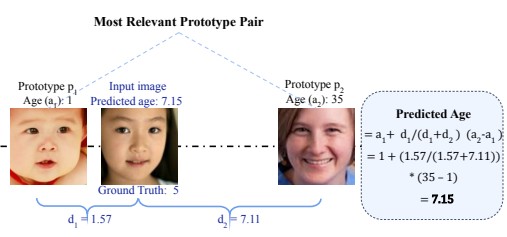

Figure 4: Reasoning process for age prediction for test image with ground-truth ages 5

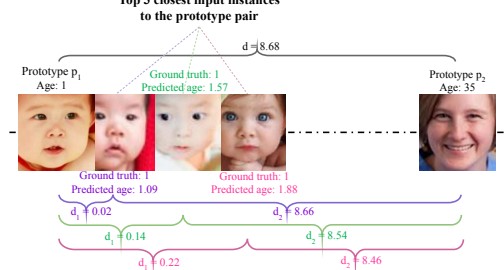

Figure 5: Global analysis for age prediction.

Figure 4 shows how ProtoPairNet predicts the age of a test image, by projecting its latent representation onto the axis defined by the most relevant prototype pair $\mathbf{p}_1$ and $\mathbf{p}_2$ selected by the pair relevance branch. The prediction branch uses the dissimilarity between the test image and the two prototypes from the most relevant pair to compute the predicted age, in this case, 7.15 for a test image with ground truth age 5. Figure 5 presents a global analysis, displaying three test instances closest to the axis defined by a given prototype pair, arranged by the predicted ages. This analysis demonstrates that the prototype pair effectively captures meaningful variations in the dataset, with the test instances positioned appropriately relative to the prototype pair along the axis. Additional examples of our ProtoPairNet's reasoning process and global analysis can be found in Appendix I and J.

### 3.2 Case Study 2: Car Racing

In this case study, we evaluate ProtoPairNet for behavior cloning in the Car Racing task from OpenAI Gym [8], using a PPO-trained expert [36]. The action space includes three continuous outputs: steering ($-1$ to $1$), acceleration ($0$ to $1$), and braking ($0$ to $1$). The dataset, generated from the expert's black-box policy, contains 36,107 states (each a stack of four frames), with 25,000 for training and the rest for validation. We evaluate ProtoPairNet as a deterministic policy deployed online in the environment. For the reported results, we use two prototype pairs per action dimension—six pairs in total across steering, acceleration, and braking. We use the same PPO configurations as Kenny et al. did in [21] to ensure a fair comparison.

We report the rewards achieved by ProtoPairNet in Table 3, comparing them to those of other regression networks. We compare ProtoPairNet with PWNet [21], which uses hand-picked prototypes, PWNet* [21] which is a version of PWNet that uses model-learned prototypes, and k-Means [21] that uses as many clusters as the number of prototypes used in PWNet. In all cases, the rewards were averaged over 5 independent runs, following the same experimental setup as described in [21] to ensure a fair comparison. The black-box expert agent achieves an average reward of 221.36, which serves as the reference point for reward comparisons. **Notably, ProtoPairNet achieves higher rewards than all other models, including the black-box expert.**

Figure 6 illustrates ProtoPairNet's reasoning process for predicting the steering action in a test scenario. The steering predicted by the PPO expert is $-0.39$, while our model predicts $-0.34$. Despite this slight difference, our model achieves a higher average reward, demonstrating its ability to outperform the expert policy while incorporating interpretability. Additional examples for reasoning process are provided in Appendix K, and global analysis examples are in Appendix L.

Table 3: Car Racing results. Comparison of different architectures in terms of rewards.

| Architecture | Reward |
|---|---|
| k-Means[21] | $-2.09 \pm 0.94$ |
| PW-Net*[21] | $-9.48 \pm 2.50$ |
| PW-Net[21] | $220.61 \pm 0.70$ |
| **ProtoPairNet** | **$223.97 \pm 0.75$** |
| PPO black box agent | $221.36 \pm 0.96$ |

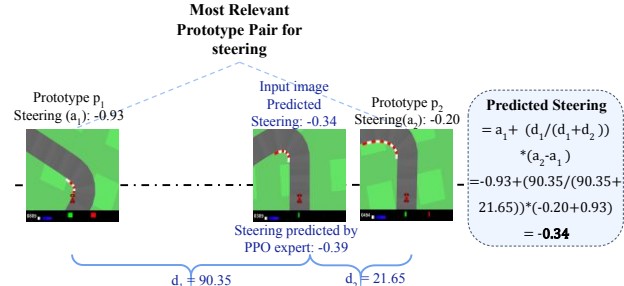

Figure 6: Reasoning process for Car Racing.

## 3.3 User Studies

We conducted four IRB-approved user studies with 120 participants (30 for each study) to evaluate the interpretability of ProtoPairNet's predictions and reasoning in Age Prediction and Car Racing tasks. In Study 1, participants viewed 10 test face images, each with two reference prototype pairs and their corresponding ages, and selected the pair that they thought better supported their age estimation. Our goal was to assess whether the prototype pairs used by our model aligned with human preferences. A majority of responses (247/300) matched the pairs selected by ProtoPairNet, indicating strong agreement with the model's selection of pairs in decision-making for age prediction. In Study 2, participants were divided into three groups and provided with 10 test images with different levels of information: Group 1 received no references, Group 2 received a single reference face with its age, and Group 3 received a pair of reference faces with their ages. We wanted to see whether providing a pair of references enhances decision-making. The MAE for age estimation was significantly lower for Group 3 (4.0) compared to Group 2 (5.96, $p < 0.01$) and Group 1 (7.85, $p < 0.01$), indicating that having a pair of references improved decision-making accuracy. For Study 3 and 4, we aimed to understand whether users found ProtoPairNet's reasoning process more preferable compared to conventional prototype-based models' reasoning that uses a single reference example. In Study 3, participants were shown applications of two reasoning processes (one using ProtoPairNet's reasoning that uses a pair of reference faces and the other using a single reference face) to 10 test images. When asked which type of reasoning they preferred, the majority (191 out of 300, $p < 0.01$) favored ProtoPairNet's. Interestingly, participants who preferred a single reference often did so when the reference closely resembled the test image, with one participant noting, *"I think when the faces look so similar to one reference, it is better to have just one reference than two."* In Study 4, participants reviewed 10 car racing scenarios, with a reasoning process that uses a pair of reference faces (ProtoPairNet) and another that uses a single reference scenario (PWNet [21]). Again, a majority (170 out of 300, $p < 0.01$) preferred ProtoPairNet's reasoning, with a pattern similar to Study 3, where those favoring single-reference reasoning did so when the test scenario closely matched the reference. These findings demonstrate that the ProtoPairNet's use of prototype pairs leads to more interpretable explanations and aligns better with human reasoning compared to existing regression models that perform one-to-one comparisons with prototypes (see Appendix A).

## 3.4 Occlusion Sensitivity Analysis

While ProtoPairNet's image-level prototypes offer intuitive, high-level interpretability (e.g., "this face looks more like that of an older person than a younger one"), they do not by themselves reveal which specific regions of an input image are most responsible for the model's prediction. To bridge this gap, we applied occlusion-based sensitivity analysis [47] to a ProtoPairNet trained for age prediction, to reveal spatial importance by systematically masking parts of an input image and measuring the impact on the model's output.

More specifically, to identify which input region the most relevant prototype pair attends to for a given input image, we sequentially masked input patches and measured the change in its prediction as an importance score. This patch occlusion method, though not built into the core architecture, provides spatial explanations without requiring part-level supervision. We analyzed the top-10 most important patches across 6 randomly chosen test images and found that high-impact regions consistently fall on key facial landmarks such as the eyes, nose, lips, and jawline (see Figure 7). This suggests that our ProtoPairNet consistently focuses on semantically meaningful features when making predictions.

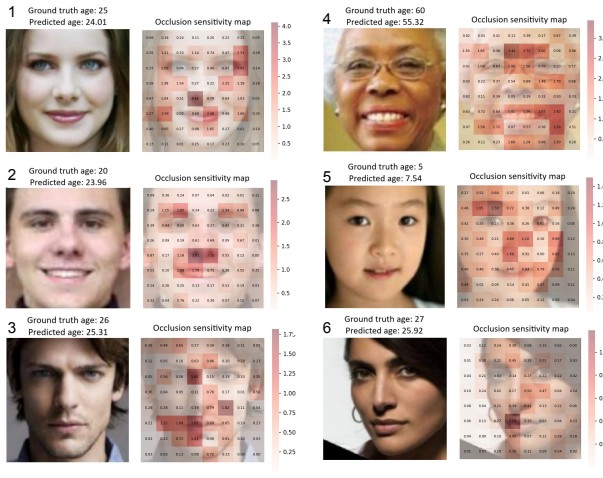

Table 4: Occlusion sensitivity results showing the average and maximum prediction changes for 6 test images.

| Test image | Average prediction change | Maximum prediction change |
|---|---|---|
| 1 | 0.773 | 4.153 |
| 2 | 0.486 | 2.904 |
| 3 | 0.376 | 1.820 |
| 4 | 0.885 | 3.438 |
| 5 | 0.371 | 1.497 |
| 6 | 0.172 | 1.192 |

Figure 7: Test images (numbered 1–6) are shown on the left, with corresponding heatmaps on the right highlighting facial regions most influential to the model's predictions.

Quantitatively, as shown in Table 4, for the 6 randomly chosen test images, the average change in prediction across occluded patches ranged from $0.172$ to $0.885$, whereas the maximum change in prediction resulting from occluding the most important patch ranged from $1.192$ to $4.153$. The substantial difference between the average and maximum prediction changes resulting from patch occlusions indicates that only a small subset of patches, typically those corresponding to key facial landmarks, has a significant impact on our ProtoPairNet's output. This further confirms the importance of these regions in our model's decision-making process.

### 3.5 Ablation Study: Is a Single, Most Relevant Prototype Pair Sufficient for a Prediction?

While our ProtoPairNet may contain multiple prototype pairs, the use of Gumbel-softmax in the pair relevance branch means that each prediction of our ProtoPairNet is essentially made by a single, most relevant prototype pair. To evaluate whether a single pair is sufficient for each prediction, we conducted an ablation study replacing Gumbel-softmax with (1) a standard softmax and (2) a simple average of all prototype pairs' predic-

Table 5: Age prediction performance with standard softmax, simple averaging, and Gumbel-softmax pair relevance.

| | MAE | $R^2$ score |
|---|---|---|
| Standard softmax | $4.49 \pm 0.007$ | $0.877 \pm 0.0006$ |
| Simple average | $4.48 \pm 0.003$ | $0.878 \pm 0.0004$ |
| Gumbel-softmax (Ours) | $4.59 \pm 0.01$ | $0.87 \pm 0.001$ |

tions. We found that using multiple pairs per prediction through standard softmax or simple averaging did not significantly improve the model's performance for age prediction, suggesting that a single, most relevant prototype pair identified by Gumbel-softmax is often sufficient for each prediction (see Table 5). More ablation studies are found in Appendix E.

## 4 Limitations, Future Work, and Conclusion

Our approach does not use part-based explanations as ProtoPNet [11] and its variants do. Leveraging prototypical parts as prototypes could improve interpretability by linking specific regions of the input to meaningful, localized concepts. However, we consider our pair-based formulation a novel contribution on its own. Our method introduces a new framework that explains predictions by positioning the latent representation of an input along a meaningful axis defined by a pair of prototypes, capturing smooth transitions and continuous variation across examples for regression tasks. Extending our framework to incorporate part-based reasoning is a promising direction we would like to explore.

To conclude, in this work, we presented the **ProtoPairNet**, a novel model for prototype-based regression that incorporates comparisons with pairs of prototypes. Our code is available at `https://github.com/Rose32/ProtoPairNet`.

## Acknowledgments

We would like to acknowledge funding from the National Science Foundation (NSF) under grants RII Track-2 FEC OIA-2218063 and NSF CAREER IIS-2442039.

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

**Appendix Table of Contents**

# A Details on User Studies

We conducted four user studies, each involving 30 participants recruited through Prolific. All participants provided informed consent prior to data collection and were fully informed of their rights. The protocols of these user studies have been approved by the IRB. Each study session lasted approximately 10 to 20 minutes, and participants were compensated at a rate of $10 per hour.

The goal of the first study was to assess whether the prototype pairs generated by our model aligned with human preferences. The second study aimed to evaluate whether presenting participants with a pair of references (i.e., a prototypical pair) would help them perform prediction tasks (e.g., age prediction) more effectively than when given only a single reference (i.e., a single prototype). Studies three and four further examined whether our model improved interpretability compared to baseline methods that used single-prototype explanations.

Overall, the results of our user studies suggest that providing pairs of explanations (prototypes) not only aligns with human preferences better, but also helps participants perform prediction tasks more effectively and provides a clearer explanation of the model's reasoning process. The following section provides further details on the examples used and the implementation of each study.

## A.1 Study 1

In this study, 30 participants were presented with 10 test photos of human faces, alongside two pairs of reference faces. These reference pairs corresponded to the training examples that our ProtoPairNet's learned prototype pairs were projected onto. For each test photo, we asked participants to choose the reference pair they felt better helped them estimate the age of the person in the photo. Our goal was to see if the prototype pairs selected by ProtoPairNet for its decision making aligned with those selected by the participants. Some example questions for Study 1 are shown in Figure 12 (a).

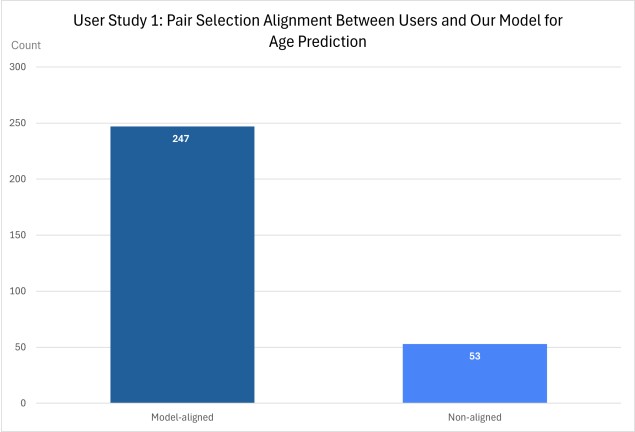

Figure 8: Study 1 result.

**Results:** The result of Study 1 is illustrated in Figure 8. We observed that the majority of users' reference pair selections were aligned with the reference prototype pairs chosen by ProtoPairNet, with 247 instances (out of 300 responses) of alignment compared to 53 instances (out of 300 responses) of non-alignment. This indicates a strong preference for the prototype pairs utilized by the model in its decision-making process for age prediction.

## A.2 Study 2

In this study, we worked with another 30 participants, dividing them into three groups of 10. Each participant was shown 10 test photos of human faces. For each test photo, participants in Group 1 were not given any reference training examples. Those in Group 2 were provided with a single reference training example, the age of the reference, and a dissimilarity score between the test photo and the reference. Participants in Group 3 were given a pair of reference training examples, their ages, and the relative dissimilarity between the test photo and the pair of references. We asked

participants to estimate the age of each test photo based on the information provided and to rate how useful they found the information in making their guesses. This study explored whether providing a pair of reference examples enhances decision-making and increases the probability of accurate predictions compared to a single example or none at all. Some example questions for Study 2 is shown in Figure 12 (b) for group 1, Figure 12 (c) for group 2, and Figure 12 (d) for group 3.

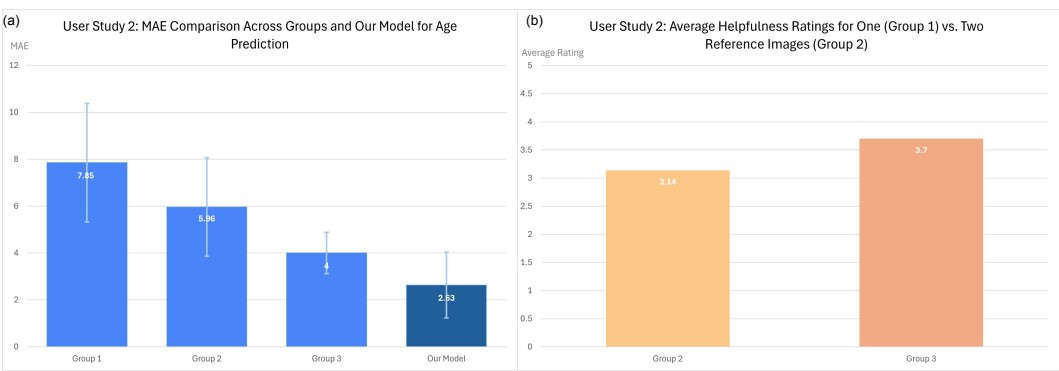

Figure 9: Study 2 results.

**Results:**  The results of Study 2 are illustrated in Figure 9. As shown in Figure 9 (a), we observed that the mean absolute error (MAE) when guessing the ages of the test photos was significantly lower for participants in Group 3 (4.0), who were provided with a pair of reference examples, compared to those in Group 2 (5.96) who were provided with a single reference example and Group 1 (7.85) who were provided with no reference example. The participants in Group 3 demonstrated better performance in predicting the age, suggesting that providing a pair of reference examples enhances decision-making accuracy. Additionally, ProtoPairNet outperformed all participant groups, achieving the lowest MAE for the test photos. To assess statistical significance, we conducted independent $t$-tests comparing Group 3 with Group 2 and Group 1. As shown in Table 6, the results indicate statistically significant differences, with Group 3 outperforming Group 2 ($t = -2.72$, $p = 0.0091$) and Group 1 ($t = -4.55$, $p = 0.0004$), confirming that the improvements observed with paired references are statistically significant.

Furthermore, as shown in Figure 9 (b), participants in Group 3 rated the helpfulness of the provided information higher on average (3.7) compared to Group 2 (3.14). This indicates that the additional context from two reference examples was perceived as more effective in aiding participants' age predictions, highlighting the importance of multiple references in improving decision-making.

Table 6: $t$-Test results comparing MAEs of Group 3 with Group 2 and Group 1.

| Groups | t-Statistic | p-Value (one-tail) |
|---|---|---|
| Group 3 − Group 2 | -2.72 | 0.0091 |
| Group 3 − Group 1 | -4.55 | 0.0004 |

### A.3 Study 3

In this study, 30 participants were shown 10 test photos of human faces. For each photo, they were presented with two reasoning processes: one using a pair of reference faces (Justification 1) and the other using a single reference face (Justification 2). We asked the participants to choose which reasoning justification they found more effective for predicting the person's age. We also added a comment box where the participants could briefly state their reasons for their choices. This study aimed to see if human users prefer the reasoning based on two reference examples or a single reference example. An example question for Study 3 is shown in Figure 13 (a).

**Results:**  The result of Study 3 is illustrated in Figure 10. The majority of participants (191 out of 300 responses) preferred the reasoning process using a pair of reference images (Justification

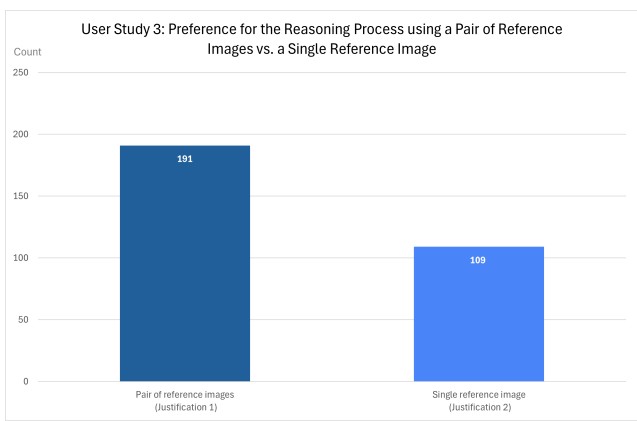

Figure 10: Study 3 results.

1) compared to fewer (109 out of 300 responses) participants who chose the process with a single reference image (Justification 2). Interestingly, based on the comments left by the participants, the cases where participants favored Justification 2 were usually when the reference face in the image closely resembled the test photo. The comment from one such participant was *"I think when the faces are closer together on one end of the spectrum it is better to have just 1 reference point."*. However, given the inherent uniqueness of human faces, relying on highly specific references is not practical and limits generalization. For the participants who favored Justification 1, they favored it strongly, leaving comments like *"Justification 1 better justifies the prediction because it compares the face with both younger and older faces, suggesting a more nuanced relationship"*, *"Justification 1 has more data points to validate its decision. Comparing two faces increases confidence compared to comparing to one face."*, etc. This considerable preference underscores the importance of the additional contextual information provided by the pair of references when predicting the ages of human faces.

The result of a one-tail $t$-test on the mean counts of the justifications chosen by the users, dubbed as the "Justification Score," further illustrate that there is a statistically significant preference for the reasoning process of using a pair of references compared to using a single reference as shown in Table 7.

Table 7: User Study 3 results comparing the Justification Score for using a pair of references and the Justification Score for using a single reference.

| Justifications | t-Statistic | p-Value (one-tail) |
|---|---|---|
| Pair of References Justification Score − Single Reference Justification Score | 5.67 | $2.34 \times 10^{-7}$ |

## A.4 Study 4

In this study, 30 participants were presented with 10 test scenarios from a car racing game. For each test scenario, participants were provided with the reasoning process our ProtoPairNet used to predict the level of acceleration (Decision 1 and Justification 1), and the reasoning process the baseline model PWNet [21] used to predict the level of acceleration (Decision 2 and Justification 2). The "Decision" here refers to the predicted acceleration and the "Justification" refers to the reasoning process of each model. Participants were asked to select the decision that made more sense to them out of the two and, separately, to choose the reasoning process they preferred. Additionally, we also put a comment box after every question for the participants to briefly justify their choices. This study aimed to understand whether users find ProtoPairNet's prediction process more intuitive and preferable compared to the baseline model's approach. An example question for Study 4 is shown in Figure 13 (b).

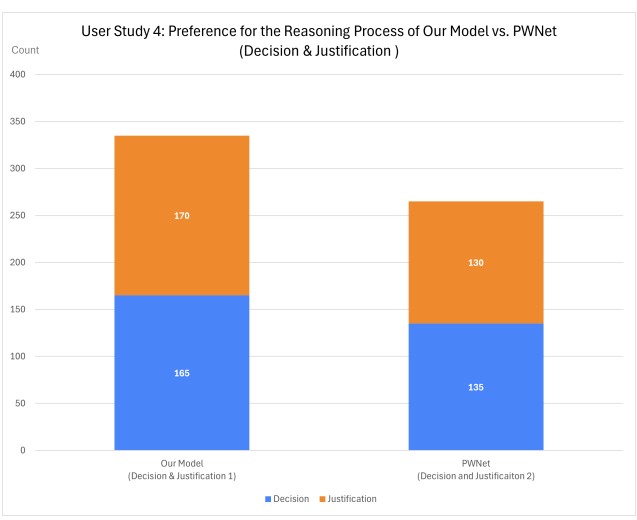

Figure 11: Study 4 results.

**Results:** The results of Study 4 are illustrated in Figure 11. We observed that the majority of participants preferred the reasoning process of ProtoPairNet (Decision and Justification 1) over the reasoning process of PWNet (Decision and Justification 2). Specifically, for decisions alone, 165 participants favored ProtoPairNet's decisions compared to 135 participants who favored PWNet's decisions. Similarly, for justifications, 170 participants preferred ProtoPairNet's justifications compared to 130 participants favoring PWNet's justifications. Based on participant feedback, those who preferred ProtoPairNet's reasoning process highlighted its clarity and the strength of the comparisons drawn, often mentioning that the reasoning process felt more comprehensive and aligned better with their expectations. As in Study 2, the participants who favored Decision and Justification 1 favored it strongly, with comments like *"It makes me nervous how Justification and Decision 2 will account for a more curved road to predict acceleration. Decision 1 considers both a curved and straight road and checks how curved the scenario is before making a decision. The way the first model makes the decision is actually good, it checks similar roads and places the sample along the spectrum and matches acceleration. The second model feels like a black box, it just says they are slightly similar and spits out a number."* This comment highlights the weakness of PWNet's manually picked prototype for acceleration which is just a straight road and is used to predict the acceleration for all scenarios in the environment. On the other hand, the participants who selected PWNet's reasoning appreciated its simplicity and as in Study 3, favored it whenever the test scenario looked significantly close to the single reference scenario provided. One such participant left the comment, *"I prefer having two frameworks of comparison, but Justification 2 having only 2% difference seems more realistic to choose"*. These results emphasize the strength of ProtoPairNet's reasoning process in both decision-making and justifications, demonstrating the added value of a robust and well-supported reasoning framework.

The result of a one-tailed $t$-test on the mean counts of justifications selected by users, or the "Justification Score" demonstrates a statistically significant preference for the reasoning process of ProtoPairNet compared to PWNet, as shown in the first row of Table 8. To further analyze participant preferences, we combine decisions and justifications into a single metric by multiplying them (i.e., Decision 1 × Justification 1 for ProtoPairNet and Decision 2 × Justification 2 for PWNet), dubbed "Preference Score" in the table. This approach highlights cases where participants consistently align with both the Decision and Justification from a model, emphasizing the importance of agreement on both aspects. The interaction between Decision and Justification is critical, as alignment with both reflects a more robust preference than agreement on either aspect individually. Again, this shows that participants found our ProtoPairNet's reasoning process more intuitive and its decisions better justified, further validating its effectiveness in this study.

Table 8: $t$-Test results comparing ProtoPairNet's scores with PWNet's scores for Justification and Preference.

| Models | t-Statistic | p-Value (one-tail) |
|---|---|---|
| ProtoPairNet's Justification Score — PWNet's Justification Score | 2.9 | 0.0026 |
| ProtoPairNet's Preference Score — PWNet's Preference Score | 2.6548 | 0.0051 |

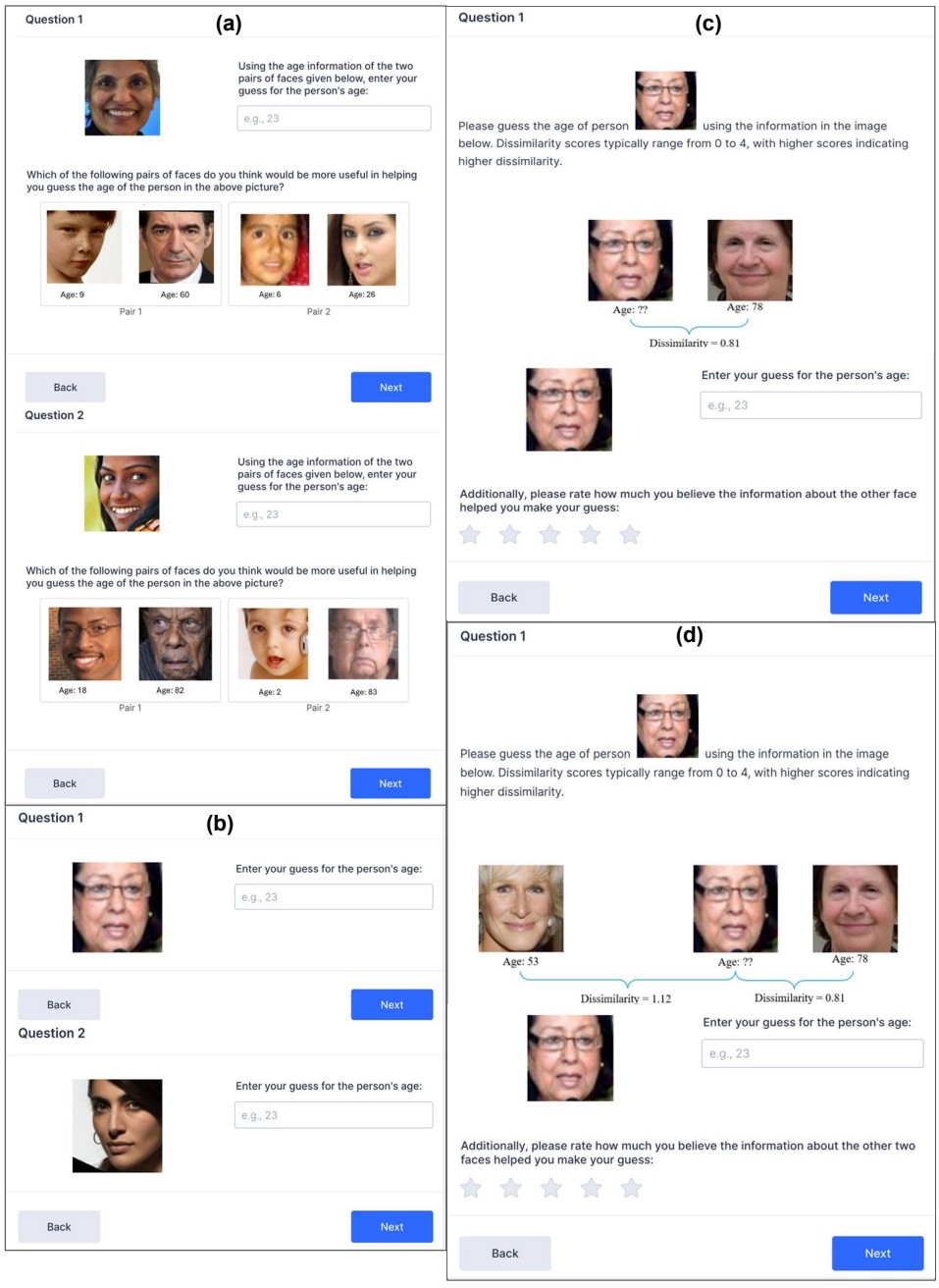

Figure 12: Example questions for Study 1 (a), and Study 2, Group 1 (b), Group 2 (c), Group 3 (d).

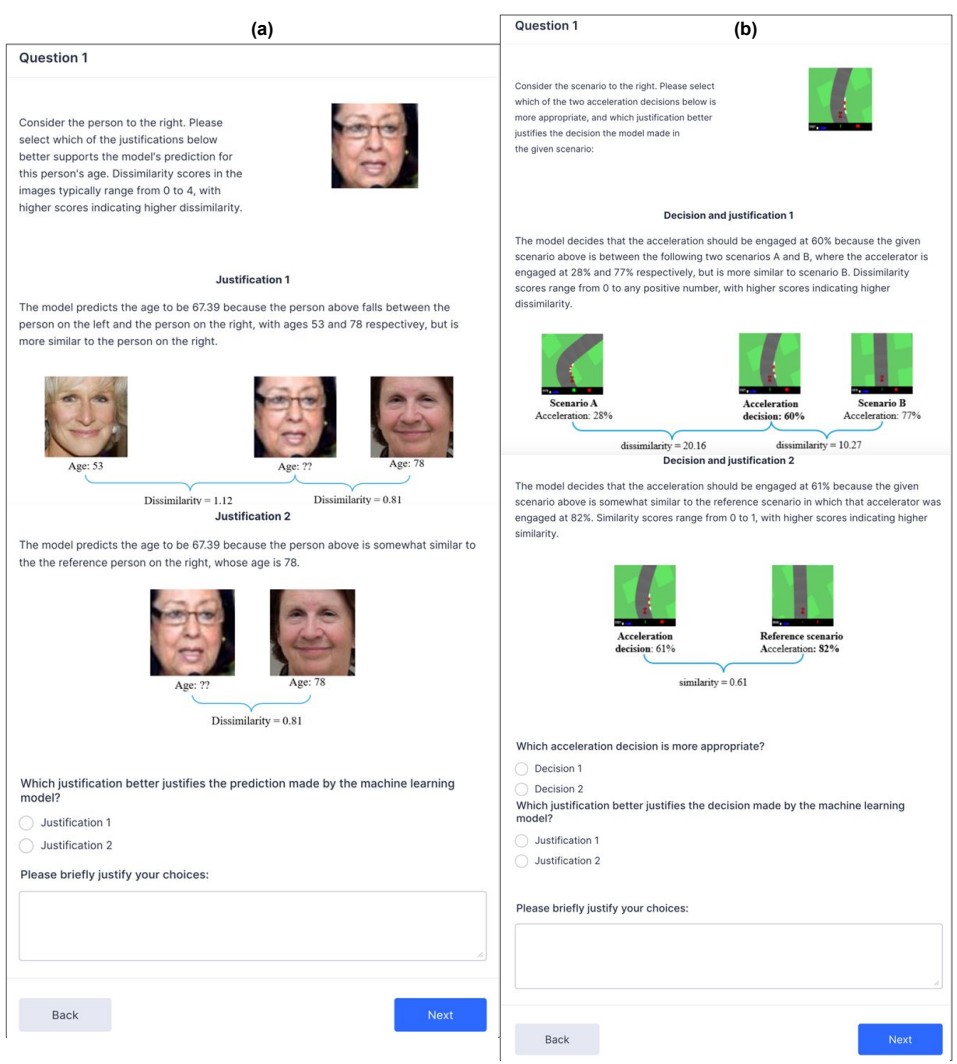

Figure 13: Example questions for Study 3 (a), and Study 4 (b).

# B  One-to-One Comparisons versus Prototype Pair-based Reasoning

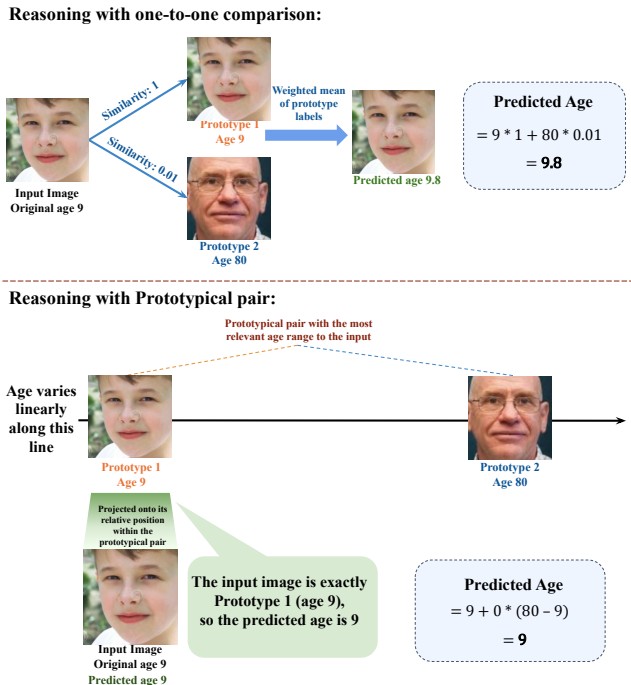

Figure 14: One-to-one comparisons versus prototype pair-based reasoning, when the input image matches a prototype exactly.

Figure 14 demonstrates why a similarity-weighted average can cause unwarranted deviations, even when an input image matches a prototype exactly. As shown at the top of Figure 14, while the input image is the same as the prototype whose label is 9, the 2-prototype model using similarity-weighted averaging predicts $9 \times 1 + 80 \times 0.01 = 9.8 \neq 9$. In contrast, the prototype pair-based reasoning (used by our ProtoPairNet) addresses this issue effectively. By design, when an input image is the same as a prototype, our ProtoPairNet predicts the label of the prototype. This is illustrated in the bottom of Figure 14.

# C  Training Hyperparameters

Table 9: Hyperparameter settings for ProtoPairNet

| Parameter | Weight |
| --- | --- |
| MAE | 1 |
| Cluster loss $\mathcal{L}_{\text{Clst}}$ | 0.008 |
| Axis distance loss $\mathcal{L}_{\text{AxisDist}}$ | 0.008 |
| Diversity loss $\mathcal{L}_{\text{Div}}$ | 0.001 |
| Gumbel softmax temperature $\tau$ | 0.1 |

This section documents the hyperparameters used to train our ProtoPairNet, as shown in Table 9. The hyperparameters shown in the table are consistent across all the architectures. The hyperparameters are chosen using grid search on a validation set for both the age prediction task and the car racing application.

To evaluate whether the Gumbel-softmax temperature $\tau$ is sufficiently small, we computed the mean $L_1$ distance between the Gumbel-softmax outputs $\mathbf{r}$ and their binarized one-hot versions $\mathbf{r}^{\text{one-hot}}$ over the test set of the age prediction task. At the temperature $\tau = 0.1$ used in our experiments, the

mean distance was 0.02, indicating that the Gumbel softmax outputs are sufficiently close to one-hot encodings.

# D    Training Procedure

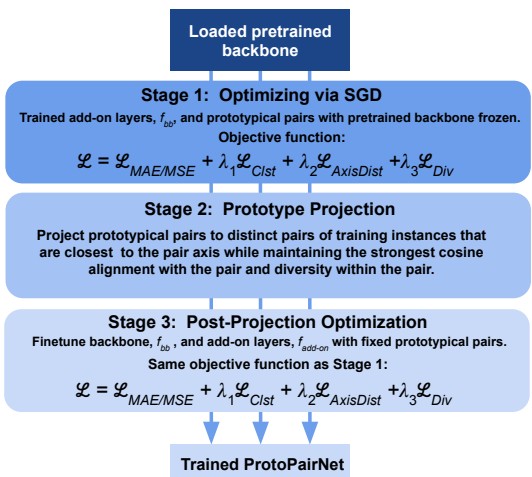

Figure 15: The procedure for training ProtoPairNet.

The three stages of training as discussed in the main paper are shown in Figure 15. For all architectures, we use backbone weights pretrained on the corresponding dataset. The encoder $f$ is composed of the backbone $f_{bb}$ and add-on layers $f_{add-on}$. In summary, this defines the basic workflow of our approach: Stage 1 trains the model along with the prototypical pairs, Stage 2 projects them onto selected training instances, and Stage 3 fine-tunes the model with a reduced learning rate while keeping the prototypical pairs fixed at the latent representations of the selected training images.

# E    Ablation Studies

The ablation study on loss terms shown in Table 10 reveals that removing the axis distance loss disrupts the projection mechanism, preventing prototypes from aligning with the most cosine-similar pair of training instances. This is evident from the fact that cosine values between prototype pairs before and after projection are no longer close to one. In contrast, removing other losses does not affect this alignment, indicating that the axis distance loss plays a crucial role in ensuring meaningful prototype placement. While the overall performance impact of removing any single loss term is not significant, these losses collectively help structure the latent space, encouraging the model to learn more useful and interpretable representations.

Table 11 and Table 12 show that increasing the prototype dimension or the number of prototype pairs does not significantly affect performance. This suggests that these hyperparameters can be adjusted flexibly based on the task without compromising model effectiveness. We find that a smaller prototype dimension is sufficient for structuring the latent space effectively, while larger dimensions do not provide additional benefits.

Table 10: Ablation study for age prediction on cluster loss, axis distance loss, and diversity loss.

| Removed parameter | MAE | $R^2$ score | Cosine alignment between prototype pairs before and after projection (5 pairs) |
|---|---|---|---|
| **None** | **4.59 ± 0.01** | **0.87 ± 0.001** | **[0.99, 0.99, 0.99, 0.99, 0.93]** |
| Cluster loss $\mathcal{L}_{\text{Clst}}$ | 4.78 ± 0.15 | 0.86 ± 0.005 | [0.99, 0.99, 0.99, 0.96, 0.93] |
| Axis distance loss $\mathcal{L}_{\text{AxisDist}}$ | 4.75 ± 0.07 | 0.86 ± 0.001 | [0.64, 0.10, 0.01, 0.73, 0.10] |
| Diversity loss $\mathcal{L}_{\text{Div}}$ | 4.83 ± 0.31 | 0.86 ± 0.010 | [0.99, 0.99, 0.96, 0.99, 0.93] |

Table 11: Effect of prototype dimension on age prediction performance.

| Prototype dimension | MAE | $R^2$ score |
|---|---|---|
| **50** (Used in paper) | **4.59 ± 0.01** | **0.87 ± 0.001** |
| 100 | 4.71 ± 0.13 | 0.87 ± 0.002 |
| 200 | 4.81 ± 0.40 | 0.86 ± 0.003 |

Table 12: Effect of the number of prototype pairs on age prediction performance. Increasing the number of prototype pairs decreases test performance. 5 pairs are sufficient and optimal in our case.

| Number of prototype pairs | MAE | $R^2$ score |
|---|---|---|
| **5** (Used in paper) | **4.59 ± 0.03** | **0.87 ± 0.001** |
| 10 | 4.85 ± 0.18 | 0.86 ± 0.004 |
| 20 | 4.89 ± 0.15 | 0.86 ± 0.007 |
| 40 | 5.24 ± 0.04 | 0.84 ± 0.001 |
| 50 | 5.58 ± 0.50 | 0.83 ± 0.019 |

## F   Car Racing Model Architecture

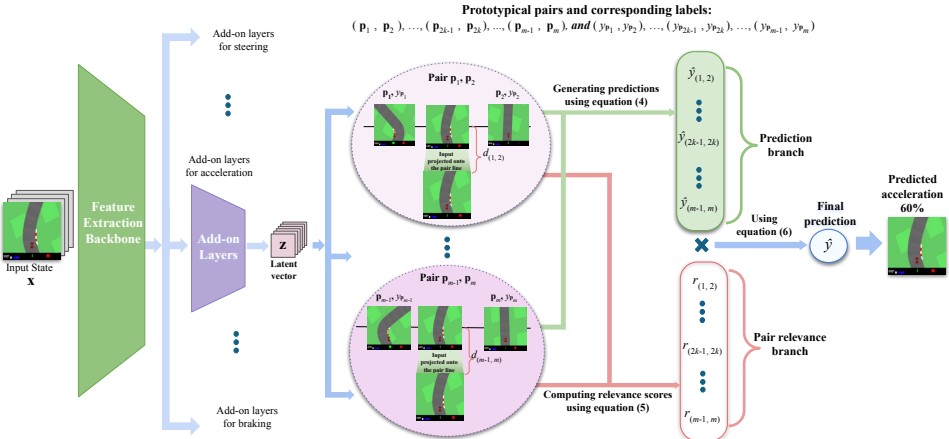

Figure 16: ProtoPairNet architecture for Car Racing. There are three separate sets of add-on layers for three output dimensions: steering, acceleration, and braking.

When the output has more than one dimension as in Car Racing (steering, acceleration, and braking), we can use a separate set of add-on layers for every dimension in our ProtoPairNet (similar to PWNet

---

**Algorithm 1** Projection of prototypical pairs

---

**Input:** Trained model $f$, training data $\mathcal{D} = \{(x^{(i)}, y^{(i)})\}_{i=1}^{n}$, prototypical pairs $\{(\mathbf{p}_{2k-1}, \mathbf{p}_{2k})\}_{k=1}^{m/2}$, prototypical labels $\{(y_{\mathbf{p}_{2k-1}}, y_{\mathbf{p}_{2k}})\}_{k=1}^{m/2}$, and empty prototypical visualizations $\{(\text{Vis}(\mathbf{p}_{2k-1}), \text{Vis}(\mathbf{p}_{2k}))\}_{k=1}^{m/2}$

**Output:** Updated prototypical pairs $\{(\mathbf{p}_{2k-1}, \mathbf{p}_{2k})\}_{k=1}^{m/2}$, labels $\{(y_{\mathbf{p}_{2k-1}}, y_{\mathbf{p}_{2k}})\}_{k=1}^{m/2}$, populated visualizations $\{(\text{Vis}(\mathbf{p}_{2k-1}), \text{Vis}(\mathbf{p}_{2k}))\}_{k=1}^{m/2}$

1: Initialize maximum cosine similarity $\alpha_k^* = 0$, maximum diversity $\beta^* = 0$, minimum distance from pairs $\gamma_k^* = \infty$ for all $k = 1, \ldots, m/2$ pairs
2: **for** each $(\mathbf{x}^{(i)}, y^{(i)}) \in \mathcal{D}$ **do**
3:     Compute latent representations $\mathbf{z}^{(i)} = f(\mathbf{x}^{(i)})$ and $d_{(2k-1,2k)}^{(i)} = \|\mathbf{z}^{(i)} - \mathbf{z}'^{(i)}_{(2k-1,2k)}\|_2^2$ for all $k = 1, \ldots, m/2$ pairs
4:     **for** $k = 1, \ldots, m/2$ **do**
5:       **for** $(\mathbf{z}^{(i)}, \mathbf{z}^{(i')}) \in$ Unique pairs **do**
6:         $\alpha_k^{(i,i')} = \frac{(\mathbf{p}_{2k-1} - \mathbf{p}_{2k})^T (\mathbf{z}^{(i)} - \mathbf{z}^{(i')})}{\|\mathbf{p}_{2k-1} - \mathbf{p}_{2k}\|_2 \|\mathbf{z}^{(i)} - \mathbf{z}^{(i')}\|_2}$
7:         $\beta^{(i,i')} = \|\mathbf{z}^{(i)} - \mathbf{z}^{(i')}\|_2^2 \cdot |y^{(i)} - y^{(i')}|$
8:         $\gamma_k^{(i,i')} = \frac{1}{2}(d_{(2k-1,2k)}^{(i)} + d_{(2k-1,2k)}^{(i')})$
9:         **if** $\alpha_k^{(i,i')} \geq \alpha_k^*$ and $\beta^{(i,i')} > \beta^*$ and $\gamma_k^{(i,i')} < \gamma_k^*$ **then**
10:           $\mathbf{p}_{2k-1} \leftarrow \mathbf{z}^{(i)}, \quad \mathbf{p}_{2k} \leftarrow \mathbf{z}^{(i')}$
11:           $y_{\mathbf{p}_{2k-1}} \leftarrow y^{(i)}, \quad y_{\mathbf{p}_{2k}} \leftarrow y^{(i')}$
12:           $\text{Vis}(\mathbf{p}_{2k-1}) \leftarrow \mathbf{x}^{(i)}, \quad \text{Vis}(\mathbf{p}_{2k}) \leftarrow \mathbf{x}^{(i')}$
13:           $\alpha_k^* \leftarrow \alpha_k^{(i,i')}$
14:           $\beta_k^* \leftarrow \beta^{(i,i')}$
15:           $\gamma_k^* \leftarrow d_{\text{avg}}$
16:         **end if**
17:       **end for**
18:     **end for**
19: **end for**
20: **Return** $\{(\mathbf{p}_{2k-1}, \mathbf{p}_{2k})\}_{k=1}^{m/2}$, $\{(y_{\mathbf{p}_{2k-1}}, y_{\mathbf{p}_{2k}})\}_{k=1}^{m/2}$, and $\{(\text{Vis}(\mathbf{p}_{2k-1}), \text{Vis}(\mathbf{p}_{2k}))\}_{k=1}^{m/2}$

---

[21]). This separates prototypical pairs of different output dimensions in different latent spaces. The model architecture in this scenario is illustrated in Figure 16.

## G   Projection of Prototypical Pairs

The pseudocode for our projection procedure is shown in Algorithm 1, with all relevant notations defined therein. We begin by initializing the cosine similarity $\alpha_k^*$ to 0, the maximum diversity $\beta^*$ to 0, and the minimum distance to the ProtoPair axis $\gamma_k^*$ to infinity. These values are tracked and updated across all batches of the training dataset. For each batch, we first compute the latent representations and their distances to all ProtoPair axes. Then, for every $k$-th prototypical pair, we search for candidate pairs of training instances that achieve the highest cosine similarity to the corresponding prototypical pair learned by the model. By doing so, we ensure that the axis formed by these candidate pairs remains well-aligned with the $k$-th ProtoPair axis. Maintaining a high cosine similarity during projection is essential to preserving the orientation of the ProtoPair axis and avoid performance degradation due to disruptive changes in direction. Mathematically, the cosine similarity $\alpha_k^{(i,i')}$ between the $k$-th ProtoPair axis and the latent direction defined by a candidate pair $\mathbf{x}^{(i)}, \mathbf{x}^{(i')}$ (whose latent representations are $\mathbf{z}^{(i)}, \mathbf{z}^{(i')}$, respectively) is given by:

$$\alpha_k^{(i,i')} = \frac{(\mathbf{p}_{2k-1} - \mathbf{p}_{2k})^T (\mathbf{z}^{(i)} - \mathbf{z}^{(i')})}{\|\mathbf{p}_{2k-1} - \mathbf{p}_{2k}\|_2 \|\mathbf{z}^{(i)} - \mathbf{z}^{(i')}\|_2}.$$

We compute this for the top 50 random samples within each training batch, restricting the search to individual batches to reduce computational cost. Next, for these candidate pairs, we compute the

diversity between the prototypes in these pairs. This is to ensure that the prototypes in the pairs exhibit sufficient stability and avoid cases where the prototypes within a pair may collapse into the same values. Mathematically, the diversity score $\beta^{(i,i')}$ between the latent representations $\mathbf{z}^{(i)}$ and $\mathbf{z}^{(i')}$, as well as the labels $y^{(i)}$ and $y^{(i')}$ of a candidate pair $\mathbf{x}^{(i)}$ and $\mathbf{x}^{(i')}$, is given by:

$$\beta^{(i,i')} = |y^{(i)} - y^{(i')}| \|\mathbf{z}^{(i)} - \mathbf{z}^{(i')}\|_2^2.$$

Finally, we compute the average axis distance $\gamma_k^{(i,i')}$ between the $k$-th ProtoPair axis and a latent candidate pair $(\mathbf{z}^{(i)}, \mathbf{z}^{(i')})$ as:

$$\gamma_k^{(i,i')} = \frac{1}{2}(\|\mathbf{z}^{(i)} - \mathbf{z}'^{(i)}_{(2k-1,2k)}\|_2^2 + \|\mathbf{z}^{(i')} - \mathbf{z}'^{(i')}_{(2k-1,2k)}\|_2^2),$$

which measures the proximity between the candidate pair and the corresponding prototypical pair in the latent space. These three metrics are used together to select the pairs that maximize cosine similarity to the corresponding prototypical pair, exhibit sufficient diversity in latent and label spaces, and have the smallest average distance to the $k$-th ProtoPair axis. We then update our model-learned prototypical pairs and their corresponding prototypical labels with these chosen pairs of latent representations and their corresponding labels. We also store the image representations or visualizations of these newly projected prototypical pairs in this stage as $\mathrm{Vis}(\mathbf{p}_{2k-1})$ and $\mathrm{Vis}(\mathbf{p}_{2k})$, where Vis denotes the prototype visualization function. This enables visualizations in the pixel space while also facilitating the next stage of training. Finally, if a candidate pair exceeds the current tracked metrics $\alpha_k^*$, $\beta^*$, and $\gamma_k^*$, we update these values to reflect the optimal candidate, as detailed in Steps 13, 14, and 15 of Algorithm 1.

## H   Age Distribution in the UTKFace Dataset

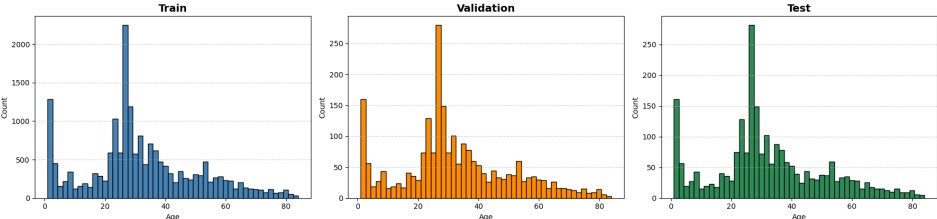

Figure 17: Age distribution across the training, validation, and test sets.

Figure 17 shows the distribution of age labels across the training, validation, and test sets. All three sets are skewed toward younger and middle-aged individuals, with relatively fewer examples at older ages. This observation provides context for the interpretability results shown in the figures in Appendix I and J. Many instances in the dataset correspond to younger individuals, which naturally align more closely with the lower end of their most relevant prototype pairs. Since our model predicts by projecting an input onto the axis of a prototypical pair in the latent space, examples from younger age groups often appear nearer to the left (younger) prototype. This behavior reflects both the data distribution and the model's reasoning.

## I   More Examples of Reasoning Process for Age Prediction

This section provides additional examples illustrating the reasoning process for age prediction. Figure 18 and Figure 19 show the reasoning process of ProtoPairNet for predicting the age of a total of 24 random input images. We find that the predicted labels of **77.22%** of test images fall within the label range of their corresponding most relevant prototype pair, while the predicted labels of the remaining **22.78%** lie outside that range. Since our ProtoPairNet makes predictions along the axis defined by a prototype pair, test samples may naturally lie either *within* or *outside* the label range of the prototype pair; both scenarios are consistent with our formulation.

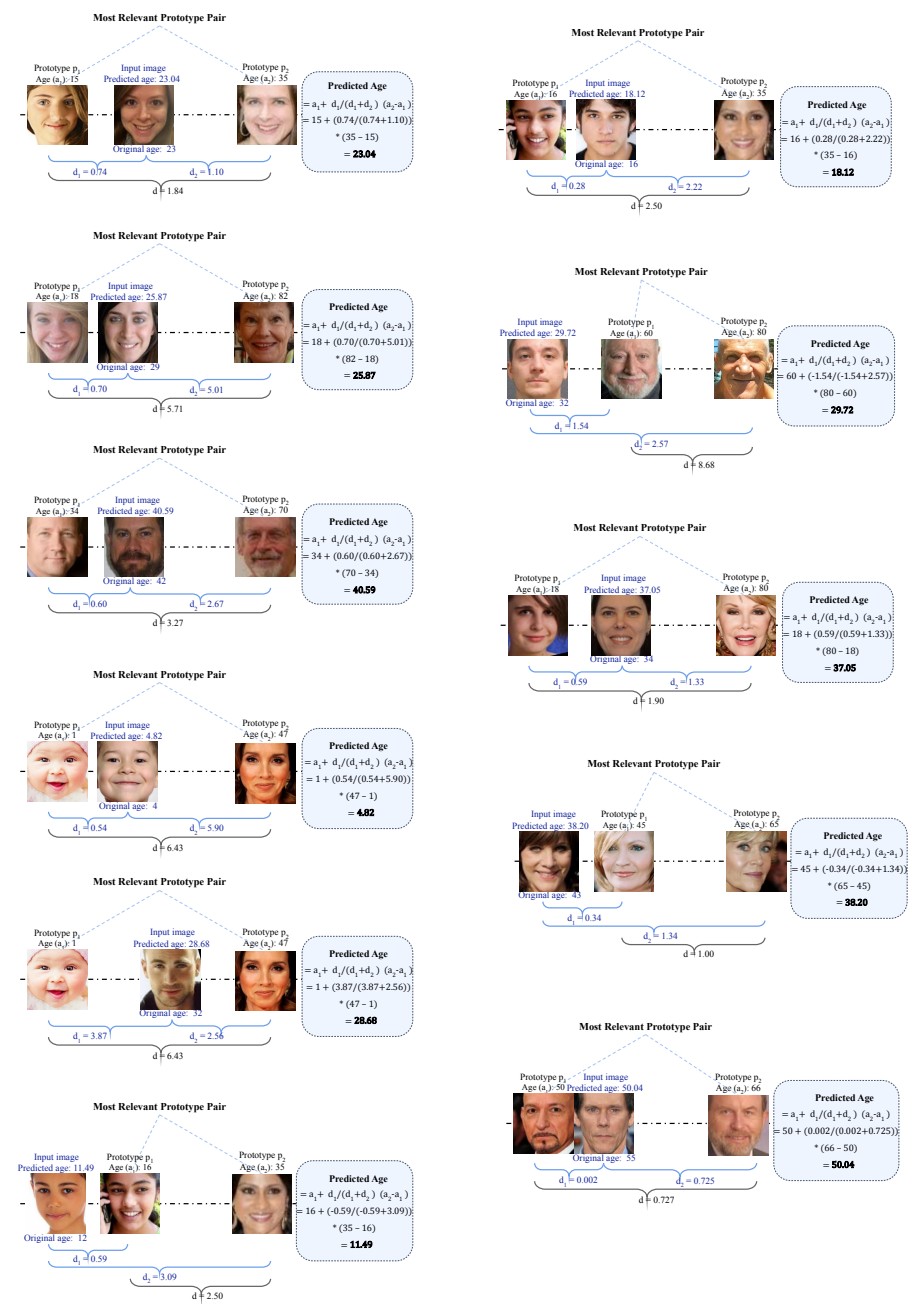

Figure 18: Example reasoning processes of how age is predicted by ProtoPairNet for 12 random input images.

# J More Examples of Global Analysis for Age Prediction

This section provides additional examples illustrating the global analysis for age prediction. Figure 20 shows the global analysis for 8 pairs of prototypes used in various ProtoPairNet models we trained.

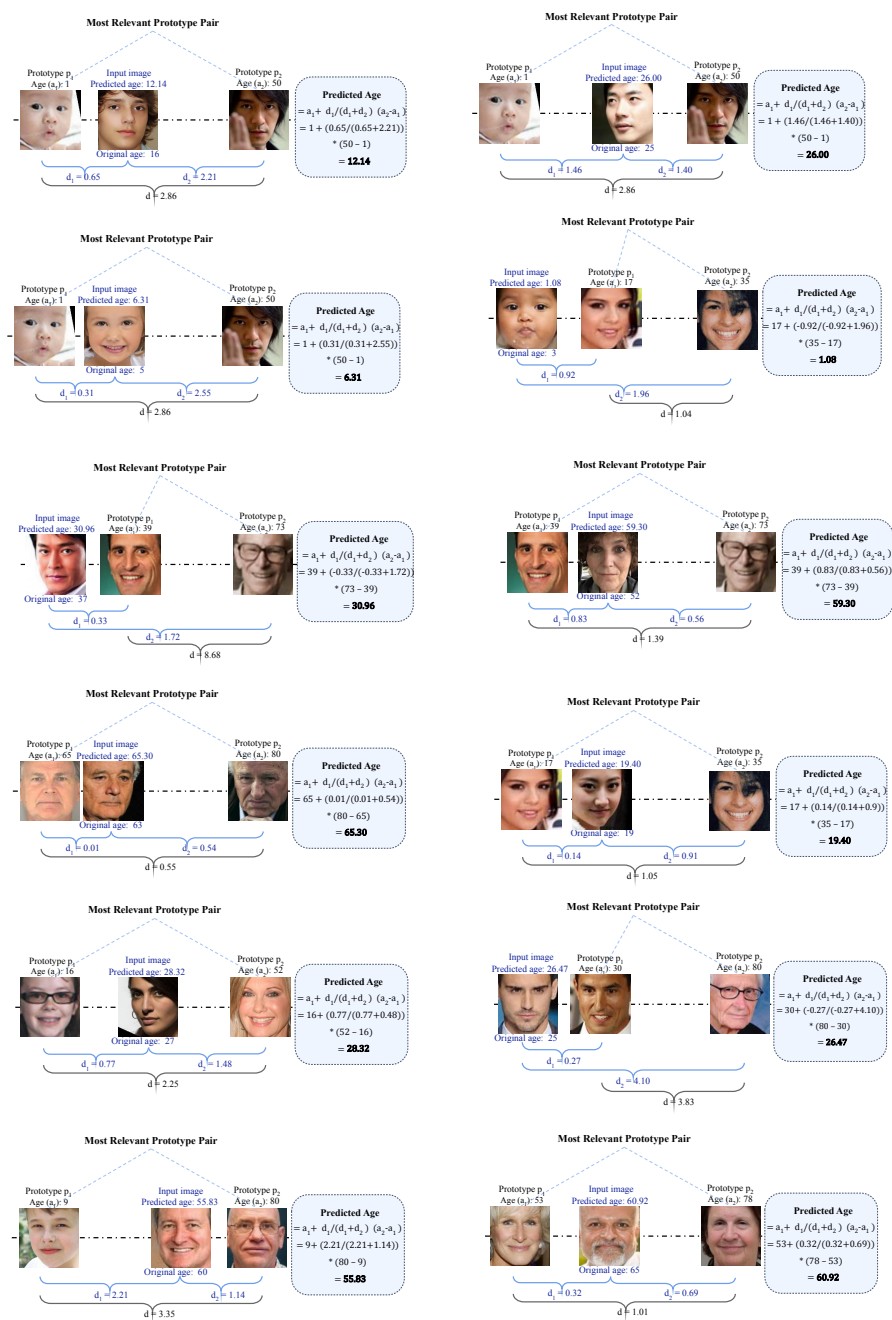

Figure 19: Example reasoning processes of how age is predicted by ProtoPairNet for 12 random input images.

# K   More Examples of Reasoning Process for Car Racing

This section provides additional examples illustrating the reasoning process for car racing. Figure 21 and Figure 22 show the reasoning process of ProtoPairNet for predicting the steering of random input states. Additionally, Figure 23 shows the reasoning process for predicting acceleration (left column) and braking (right column).

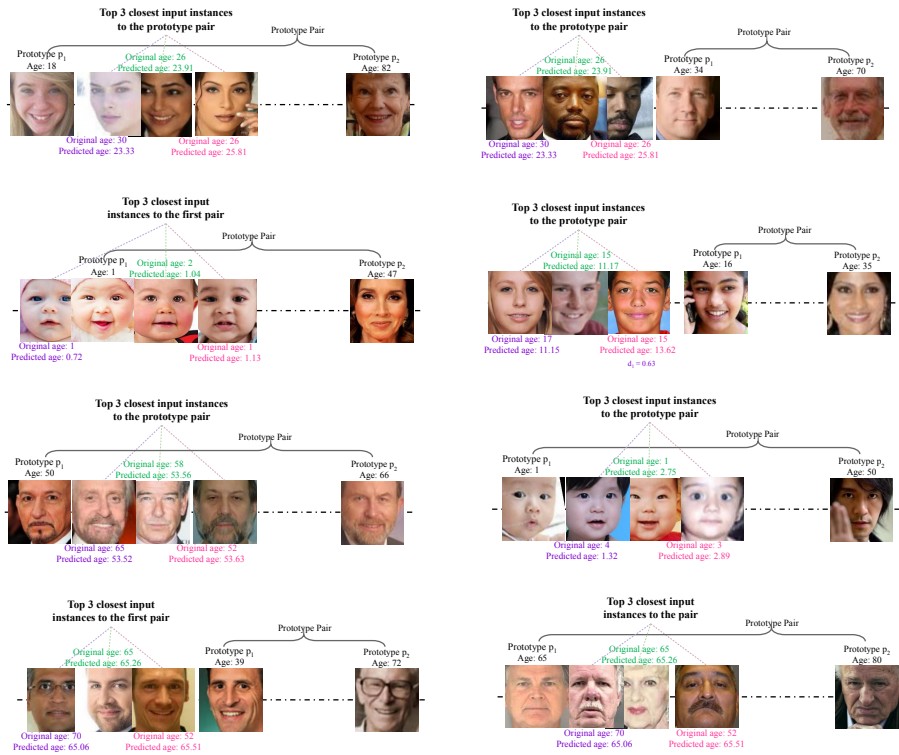

Figure 20: Global analyses of 8 pairs of prototypes used in ProtoPairNet.

## L   More Examples of Global Analysis for Car Racing

This section provides additional examples illustrating the global analysis for car racing. Figure 24 shows the global analysis for prototype pairs for steering. Additionally, Figure 25 shows the global analysis for a pair of acceleration prototypes (left column) and a pair of braking prototypes (right column) used by ProtoPairNet.

## M   Reproducibility and Empirical Computational Cost

We implemented our models using PyTorch and conducted all experiments on a high-performance computing cluster using SLURM. Each experiment was run on a single NVIDIA A100 80GB PCIe GPU with CUDA version 12.3, using 2 CPU cores and 64 GB of memory. The full pipeline—including prototype projection and fine-tuning—took approximately 6 hours for age prediction (3 runs, batch size 256) and 3 hours for car racing (5 runs, batch size 128). Prototype projection added minimal computational overhead in both tasks. For comparison, HPN [24] took approximately 1.5 days (36 hours) to complete 3 runs of 250 epochs using their default configuration, as reported in their paper. INSightR-Net [16] required approximately 5 hours, while the ExPeRT [17] required just under 2.5 hours to train under similar conditions.

## N   Broader Impacts

Model interpretability is essential for the deployment of artificial intelligence (AI) systems in high-stakes environments. This work advances interpretability for tasks requiring continuous target predictions by leveraging prototypical pairs that ground predictions in semantically meaningful examples. By structuring predictions around these visual anchors, our method enables users to understand not only what the model predicts, but also why. Our approach supports applications where trust, transparency, and accountability are crucial, such as medical decision support, autonomous driving, and human-centered AI, and offers a path toward more interpretable and human-aligned AI systems.

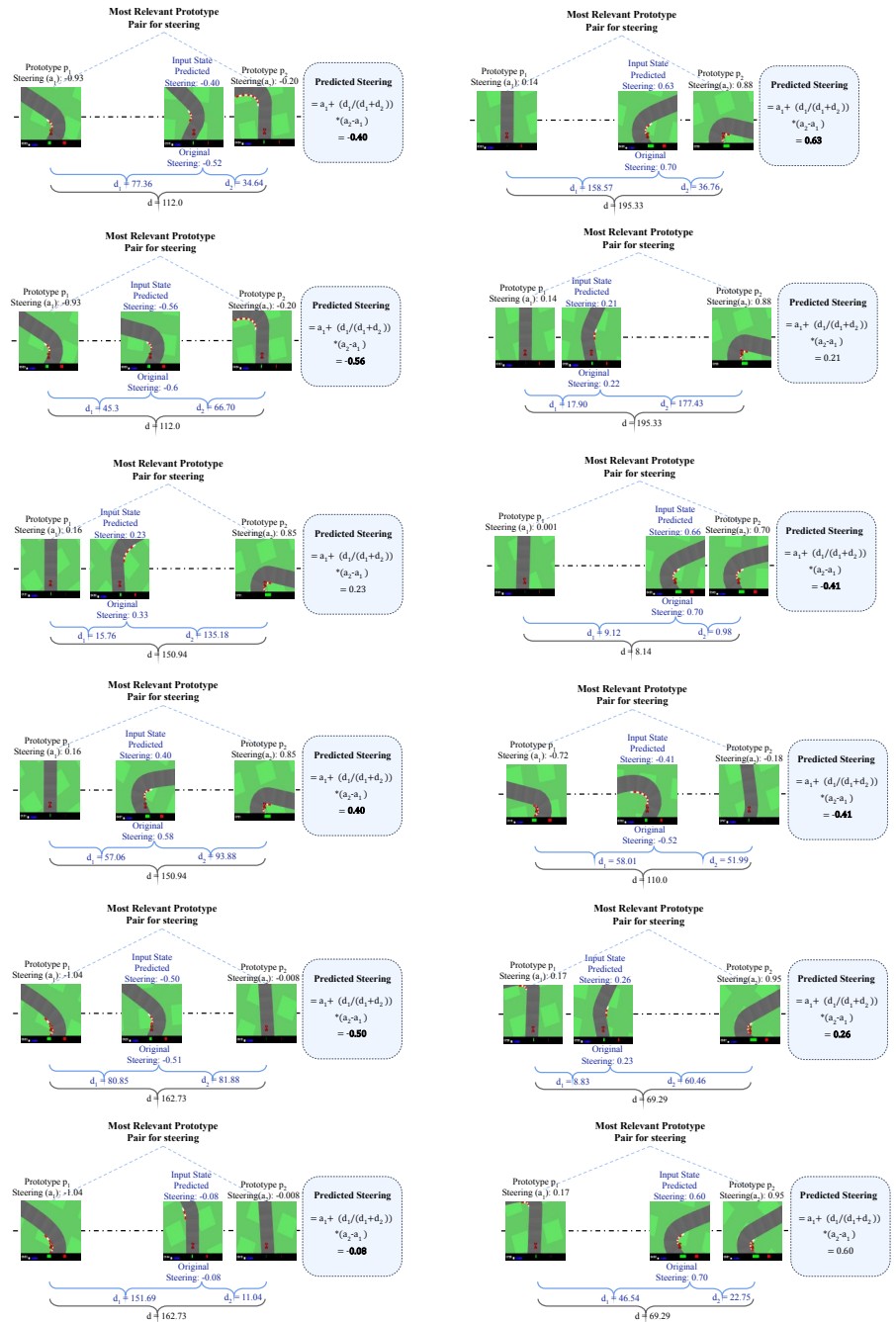

Figure 21: Example reasoning processes of how steering is predicted by ProtoPairNet for 12 random input states.

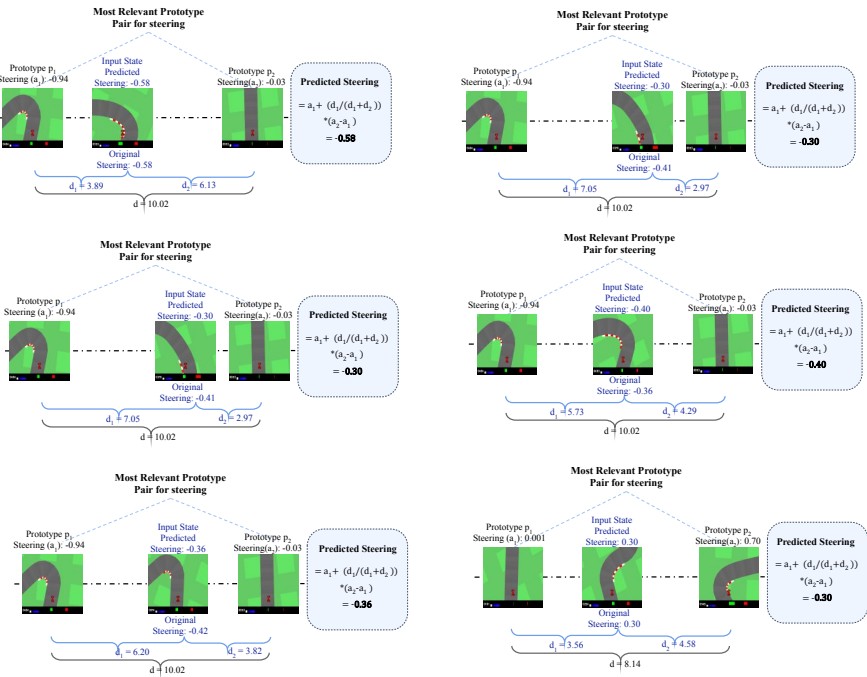

Figure 22: Example reasoning processes of how steering is predicted by ProtoPairNet for 6 random input states.

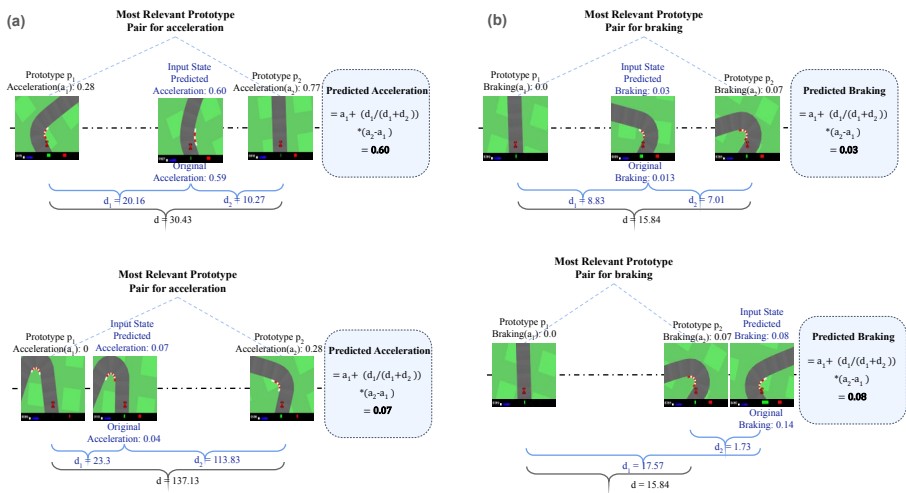

Figure 23: Example reasoning processes of how acceleration is predicted for 2 random input states, left column (a), and how brake is predicted for 2 random input states, right column (b) by ProtoPairNet.

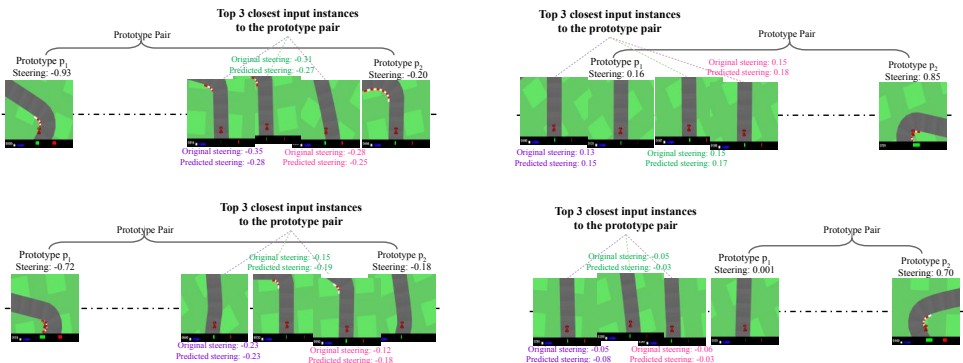

Figure 24: Global analyses of 4 pairs of prototypes used by ProtoPairNet for steering in car racing.

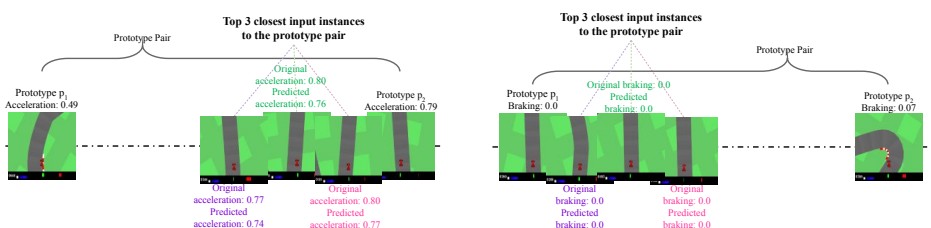

Figure 25: Global analyses of a pair of prototypes for acceleration, left column (a), and a pair for braking used by ProtoPairNet for car racing.

