# OpenReview forum: "ProtoPairNet: Interpretable Regression through Prototypical Pair Reasoning"
_NeurIPS.cc/2025/Conference — NeurIPS 2025 poster_

### Official Review · Reviewer_pNkw · 2025-06-20

**Clarity:** 3
**Significance:** 3
**Originality:** 3
**Rating:** 5
**Confidence:** 4

**Summary:**

The authors propose ProtoPairNet as an interpretable approach to regression through prototypical pair reasoning. Experiments show the utility of the proposed approach in both computer vision and reinforcement learning settings. Several human studies are conducted that further highlight the benefits of the pair based approach.

**Questions:**

- Can you provide some more clarity on the assumptions of the prediction branch? It would be nice to know of scenarios when these assumptions would break so I can better understand the assumptions themselves. Would the labels not always vary linearly along the k-th ProtoPair axis?

- Is one prototype from the pair always on either side of the projected input image?

- What is the motivation for the diversity criterion? Do you not want the prototypes to be always similar to the input image? Was training quite difficult given the number of terms in the loss function?

- What were the limitations of the user study?

- Why do you not use a part-based framework, given the success of this approach in other works.

**Ethical Concerns:**

["NO or VERY MINOR ethics concerns only"]

**Final Justification:**

From the rebuttal the authors have improved the quality and clarity of the manuscript. I think this is a good paper offering a novel solution to a problem with limited coverage in the literature (prototypes for regression). One notable weakness in my opinion is in the user study.

**Limitations:**

- Authors could go into more detail about the computational advantages or drawbacks of the proposed approach. There is related work on prototypes and criticisms, I also wonder if using maximum mean discrepancy would have advantages over the k-means based approach used here?

- Not using a part-based framework is a notable weakness. The authors mention this as a weakness in the limitations but I think some rationale is needed here given that most developments in the field recognise the benefits of part-based approaches.

- Authors mention code upon acceptance but I think it would actually have been useful to provide the anonymised code in the supplement or on one of the anonymous archives as this would give further clarity on methodological approaches.

- User study design (see weaknesses)

**Paper Formatting Concerns:**

No major concerns here.

**Quality:**

3

**Strengths And Weaknesses:**

Prototype based approaches are uncommon in regression scenarios and solutions would be a welcome and useful contribution. Modernising case-based solutions are one of Rudin’s 10 Grand Challenges for interpretable machine learning.

The approaches and comparisons are reasonably well grounded in the literature. However, there is no discussion of seminal work on prototypes and criticisms [1]. This is especially relevant as the user study in this prior work also shows the benefits of using a prototype and another example (a criticism in the case of [1]).

There is a theoretical contribution for both prediction and pair relevance. The design is (for the most part) intuitive and I liked the geometric interpretation.

While the inclusion of a user study is important, there are several and design questions and flaws. How was the number of participants determined? It does not seem like there was a power analysis conducted to determine this? It does not seem like the experiment was controlled for the number of items presented to users. Would it not have made more sense to compare a prototypical pair against either/both (i) a single prototype and a random prototype (ii) two random prototypes. The problem is that with the current design, the number of items presented to users is not fixed and could therefore be considered a confounding variable. Were there attention checks? This is typically considered best practice in user study design. The figures in the appendix that show the set-up (e.g., figure 11) should probably go in the main text for clarity. In fact, the appendix was quite a big document but it was rarely referred to in the main text.


[1] Kim, B., Khanna, R. and Koyejo, O.O., 2016. Examples are not enough, learn to criticize! criticism for interpretability. Advances in neural information processing systems, 29.

---

> ### Author Rebuttal · Authors · 2025-07-31
>
> - No discussion of Kim et al., 2016 (Strengths and Weaknesses paragraph 2):
>
> Thank you for highlighting this important prior work. While Kim et al. (2016) [1] shares our motivation of improving interpretability through comparison, their prototype + criticism framework is designed for classification, not regression. In contrast, ProtoPairNet introduces prototype pairs to define semantic axes for continuous-valued prediction, enabling structured reasoning in regression tasks. We agree the work is related in spirit and will include a discussion of it in the final version.
>
> - User study design (Strengths and Weaknesses paragraph 4, Question 4, Limitation 4):
>
> We chose a sample size of 30 participants based on practical considerations and methodological precedent. A sample of 30 is widely accepted as a minimum for the Central Limit Theorem to apply [2, 3].
>
> Number of items shown to users: You are right that controlling the number of items shown is important to avoid confounds. However, our primary goal was to compare prototype pair reasoning with individual prototype-based explanations, which naturally led to different numbers of items being shown.
>
> That said, in Study 1, we conducted a controlled comparison where users were shown two prototype pairs—one used by the model and one distractor—and asked to guess the target image’s age and choose the more helpful pair. Users largely agreed with the model’s choice, supporting the effectiveness of prototype-pair reasoning even under controlled conditions.
>
> Attention checks: Although we did not include explicit attention checks, we flagged and removed responses from users with abnormally short completion times and replaced them with new participants. In future studies, we plan to add dedicated attention checks to improve data quality.
>
> ​​We agree that including study setup figures like Figure 11 in the main text would improve clarity, and we will try to incorporate them in the final version within page limits. We will also ensure better referencing of relevant appendix sections.
>
> - Can you provide more clarity on the assumptions of the prediction branch? (Question 1)
>
> The prediction branch of ProtoPairNet relies on two key assumptions:
>
> 1. The label is assumed to vary linearly along the axis defined by a prototype pair in latent space.
>
> 2. Inputs with the same target value are assumed to lie on a common hyperplane orthogonal to the prototype pair axis.
>
> These assumptions are directly built into Equation (4) of our paper, which defines the prediction made by a prototype pair. Importantly, the two assumptions are not imposed directly on the raw data, but rather on the latent space produced by the model’s backbone. A well-trained ProtoPairNet is optimized to shape the latent space such that these assumptions hold approximately true.
>
> If the assumptions were violated – e.g., if labels were not linearly aligned with prototype axes – prediction accuracy would suffer, resulting in high MAE and poor R². Since ProtoPairNet achieves strong test performance, the assumptions appear to be satisfied in practice.
>
> - Is one prototype from the pair always on either side of the projected input image? (Question 2)
>
> No. The input image's latent representation need not fall between both prototypes. In the bottom-left example of Figure 17 in Appendix I, the model predicts an age of 11.49 for an individual of age 12, using a pair of prototypes aged 16 and 35. Here, the input lies to the left of both prototypes in latent space. More such cases appear in Figures 17 and 18.
>
> - What is the motivation for the diversity criterion? Do you not want the prototypes to be always similar to the input image? Was training quite difficult given the number of terms in the loss function? (Question 3)
>
> The diversity criterion ensures that each prototype pair captures a meaningful direction in the latent space. While some prototypes should resemble the input (via cluster loss), the diversity criterion prevents pairs from being too similar, thereby avoiding poorly defined axes and supporting a more meaningful interpolation. Despite multiple loss terms, training was stable and converged reliably using our multi-stage procedure (Section 2.4).
>
> - Why not use a part-based framework? Not using a part-based framework is a notable weakness (Question 5, Limitation 2)
>
> Prior work on prototype-based deep neural networks spans two major approaches: using prototypes defined at the global (entire image) level [4, 5] and those defined at the local (patch or part) level [6]. Our choice to adopt image-level prototypes aligns with the former line of work and is a deliberate design decision, primarily due to the nature of our data and labels. Specifically, for age predictions, learning semantically meaningful part prototypes can be challenging, since a 50-year-old may have facial features that resemble someone younger, making part-based prototypes less reliable.
>
> That said, we fully agree with the concern about not using a part-based framework. Fortunately, this limitation can be addressed by post-hoc attribution methods. For instance, occlusion-based sensitivity analysis [7] or input perturbation methods [8] can help localize which input regions most influence the similarity (or dissimilarity) to a given prototype pair. This approach has precedent: Ragodos et al. [5] also used global prototypes (entire game frames) but applied post-hoc attribution to identify salient regions that drive prototype similarity.
>
> In our setting, to identify which input region a prototype pair attends to for a given input image, we sequentially masked input patches and measured the change in its prediction as an importance score. This patch occlusion method, though not built into the core architecture, provides spatial explanations without requiring unreliable part-level supervision. We analyzed the top-10 most important patches across 5 randomly chosen test images and found that high-impact regions consistently fall within rows 2–6 and columns 2–6 of the 8×8 patch grid—corresponding to key facial landmarks such as the eyes, nose, lips, and jawline. This suggests the model consistently focuses on semantically meaningful features.
>
> Quantitatively, as shown in the table below, the average change in prediction across the occluded patches ranged from 0.38 to 1.07, with maximum change for each image between 1.82 and 4.40, confirming that these regions exert a significantly greater influence on the model’s output. In all cases, these regions corresponded to one of the facial landmarks.
>
> | Test image   | Average prediction change | Max prediction change |
> |---------------|----------------|-------------|
> | Test image 1  | 0.525  | 2.737 |
> | Test image 2  | 0.773  | 4.153 |
> | Test image 3  | 0.378  | 1.820 |
> | Test image 4  | 1.071  | 4.400 |
> | Test image 5  | 0.884  | 3.438 |
>
> - Computational advantages or drawbacks (Limitation 1):
>
> While ProtoPairNet does introduce some computational overhead due to the need to learn and maintain prototype pairs in latent space and the multi-stage training procedure, in practice, we found that its training time is shorter than, or similar to, that of other existing models (Appendix M). In particular, we have found that ProtoPairNet training is much faster than HPN [9] training.
>
> In addition to enhanced interpretability, ProtoPairNet offers several computational advantages:
> ProtoPairNet’s reasoning based on Equations (4) and (6) can be implemented efficiently using existing deep learning frameworks (e.g., PyTorch).
> ProtoPairNet uses a fixed set of prototype pairs, rather than relying on large nearest-neighbor databases or exemplar memory, which reduces memory requirements and lookup time.
> The use of Gumbel-softmax allows differentiable, sparse, and low-cost identification of the most relevant prototype pairs.
>
> - Maximum mean discrepancy (MMD) versus k-means based approach (Limitation 1):
>
> MMD [1] offers distributional matching but is computationally heavier and less aligned with our geometrical setup. Our k-means-based approach allows explicit control over prototype diversity and easily grounds prototype pair axes in real data pairs. It also scales efficiently with latent dimensions and dataset size.
>
> - Code availability (Limitation 3):
>
> Anonymized code was uploaded to OpenReview as a supplementary material (in a .zip file) prior to the submission deadline. Please download the supplementary material of our paper.
>
> [1] Kim, B., Khanna, R. and Koyejo, O.O., 2016. Examples are not enough, learn to criticize! criticism for interpretability. Advances in neural information processing systems, 29.
>
> [2] Zhang, Xijuan, et al. "How to think clearly about the central limit theorem." Psychological Methods 28.6 (2023): 1427.
>
> [3] Matz, David C., and Emily L. Hause. "“Dealing” with the central limit theorem." Teaching of Psychology 35.3 (2008): 198-200.
>
> [4] Kenny, E. M., Tucker, M., & Shah, J. (2023). Towards interpretable deep reinforcement learning with human-friendly prototypes. In The Eleventh International Conference on Learning Representations.
>
> [5] Ragodos, R., Wang, T., Lin, Q., & Zhou, X. (2022). Protox: Explaining a reinforcement learning agent via prototyping. Advances in Neural Information Processing Systems, 35, 27239-27252.
>
>
> [6] Chen, C., Li, O., Tao, D., Barnett, A., Rudin, C., & Su, J. K. (2019). This looks like that: deep learning for interpretable image recognition. Advances in neural information processing systems, 32.
>
> [7] Zeiler, M. D., & Fergus, R. (2014, September). Visualizing and understanding convolutional networks. In European conference on computer vision (pp. 818-833). Cham: Springer International Publishing.
>
> [8] Petsiuk, V., Das, A., & Saenko, K. (2018) RISE: Randomized Input Sampling for Explanation of Black-box Models. British Machine Vision Conference.
>
> [9] Mettes, P., Van der Pol, E., & Snoek, C. (2019). Hyperspherical prototype networks. Advances in neural information processing systems, 32.

---

> > ### Comment · Reviewer_pNkw · 2025-08-04
> > **Thanks**
> >
> > Thank you for the response to my concerns. Including these details in the final manuscript would improve both the quality and clarity of the paper. I do think the limitations of the user study should be discussed and flagged, especially in light of very recent position papers such as [1].
> >
> > Taking this into account, I have increased my scores. Good luck!
> >
> > [1] Pičulin, Matej, et al. 2025 "Position: Explainable AI Cannot Advance Without Better User Studies." Forty-second International Conference on Machine Learning Position Paper Track.

---

> ### Author Response · Authors · 2025-08-07
> **Thank you for your feedback**
>
> Thank you for your thoughtful feedback and for taking the time to review our rebuttal. We appreciate your willingness to increase our paper's scores. We will incorporate details from our rebuttal into the final paper to improve its quality and clarity, and we will also include a discussion of the limitations of the user studies.

---

### Official Review · Reviewer_gJBu · 2025-07-01

**Clarity:** 2
**Significance:** 2
**Originality:** 3
**Rating:** 4
**Confidence:** 3

**Summary:**

This paper proposes **Prototypical Pair Network (ProtoPairNet)**, a novel model belonging to the class of prototype-based models, which offers the promise of interpretability versus standard black-box models.

ProtoPairNet tackles the regression problem instead of the classification problem, which has been the dominant focus of the community. While some prior works have explored regression with prototype-based models, this paper argues that those approaches suffer from one of the following limitations: (1) they rely on similarity-weighted averaging, which introduces interference from unrelated prototypes, or (2) they sacrifice interpretability.

To address this, the paper introduces two key innovations:

(1) Instead of using single instances as prototypes, the model uses **pairs** of instances, such that the regression output varies linearly along the line connecting the two instances.

(2) Given a test input, the model computes relevance scores with respect to a library of prototype pairs, selects one using a soft mechanism (Gumbel-Softmax), and applies a linear interpolation formula to compute the output from the labels of the selected prototype pair.

In experiments, the method performs competitively with black-box models and outperforms other previous prototype-based models. User studies also suggest that the model’s reasoning process aligns with that of human participants.

**Questions:**

## Questions

Q. Why is now a timely moment for this paper? All the ingredients motivating this work seem to have existed already a few years ago.

Q. How are prototype pairs decided? How is the pairing done?

Q. I tend to believe that interpretability is usually gained at the expense of flexibility, so one would expect prototype models (which are interpretable) to perform worse than the black-box models -- which isn't a bad thing, but just something I would expect. However, Table 1 suggests the opposite, and also, it appears that ProtoPairNet even outperforms black-box PPO. I was wondering if the authors have an explanation about this? Would including additional benchmarks help here?

Q. Have the authors tried using any pre-trained backbones?

Q. If the perpendicular drop from the input latent to the prototype axis lies outside the segment formed by the prototype pair, how is this case handled?

## Suggestions for Improvement

An ablation of the three “metrics” and the various loss terms would be informative.

**Ethical Concerns:**

["NO or VERY MINOR ethics concerns only"]

**Final Justification:**

My main initial concerns were around things that were unclear to me (mainly because of their writing quality in my opinion), e.g., it was not quite clear if and how prototypes were decided/learned. Even though I understand these points better now after reading the rebuttal response, and therefore I am increasing the score, I am broadly concerned that the quality of writing in the paper could be made cleaner if the community has to derive the full value out of this paper.

**Limitations:**

yes

**Paper Formatting Concerns:**

None.

**Quality:**

2

**Strengths And Weaknesses:**

## Pros

1. The paper is well-written and easy to follow.
2. The paper is appropriately scoped.
3. The idea of using prototype pairs is novel.

## Cons

1. The motivation for using pairs versus single prototypes is somewhat unclear. The issue of interference from irrelevant prototypes could be addressed using Gumbel-Softmax alone. So, what necessitates the use of *pairs* of prototypes?
2. If the intent is to increase expressiveness, it would be nice to have an ablation that varies the tuple size from 1 to 2 and to perhaps even higher-order tuples. Additionally, the following aspects are unclear to me:

    (1) How are the pairs chosen to form the prototypes?

    (2) Are these prototypes decided during training, or are they provided at test time?

    (3) Does the method explicitly enforce the “linearity” of the target function with respect to the latent representations of the prototype pairs?


See also the “Questions”.

---

> ### Author Rebuttal · Authors · 2025-07-31
>
> - Motivation for using pairs (Cons 1):
>
> While Gumbel-softmax helps with sparse prototype selection, it does not address the core limitation of using single prototypes for regression. Specifically, a one-to-one comparison between an input and a single prototype only tells us that the input is “similar” to that prototype, and therefore its label should be similar. However, in regression, this is not sufficient – we often need to know in which direction the output should differ (i.e., should it be greater or smaller than the prototype's label). As shown in Figure 1 of our paper, reasoning along a prototype pair provides this directional context: the model can predict where along the axis between two known values the input lies. This cannot be achieved with reasoning using single prototypes.
>
> - Ablation study for varying number of prototype pairs (Cons 2):
>
> Below, we include an ablation study on the number of prototype pairs used in ProtoPairNet. Results for 5 and more pairs are also provided in Appendix E. As shown in the table, performance differences are minimal for 5 pairs or fewer, but we chose 5 pairs to allow greater expressiveness. Compared to using only 1 or 2 pairs, having 5 allows the model to represent a wider range of prototypical variations (e.g., across age groups or demographics), making the reasoning behind predictions more accessible to human understanding.
>
> | Number of prototype pairs | MAE           | R² score      |
> |-----------------------------|---------------|---------------|
> | 1 pair  | 4.50 ± 0.02  | 0.877 ± 0.002 |
> | 2 pairs | 4.51 ± 0.01  | 0.878 ± 0.0005|
> | 5 pairs | 4.52 ± 0.01  | 0.876 ± 0.0009|
> | 10 pairs | 4.85 ± 0.18  | 0.86 ± 0.004|
> | 20 pairs | 4.89 ± 0.15  | 0.86 ± 0.007|
> | 40 pairs | 5.24 ± 0.04  | 0.84 ± 0.001|
> | 50 pairs | 5.58 ± 0.50  | 0.83 ± 0.019|
>
> - How are the prototype pairs chosen? Are they chosen during training? How is the pairing done? (Cons 2(1), Cons 2(2), Question 2)
>
> Prototype pairs are selected during Stage 2 of our training algorithm (see pages 6–7 of the paper). Before this stage, prototype pairs exist as abstract vectors in the latent space and are not tied to specific training examples. In Stage 2, we replace each abstract prototype pair with a pair of training images chosen based on the following three criteria:
>
> (i) Cosine similarity – the candidate pair should define an axis closely aligned with the original abstract prototype pair;
>
> (ii) Diversity – the two images in the pair should be sufficiently far apart in the latent space to capture meaningful variation;
>
> (iii) Average axis distance – the candidate pair’s axis should not deviate too much from the current prototype pair axis.
>
> For each prototype pair, the algorithm searches the training set for a pair of images that maximizes cosine similarity (alignment) while satisfying minimum diversity and maximum average axis distance constraints. Once selected, we update the prototype pair to the latent representations of the chosen images and assign their ground-truth labels as the new prototypical labels. More details can be found in Section 2.4 of the paper.
>
> - Does the method enforce “linearity” of the target function with respect to the latent representations of the prototype pairs? (Cons 2(3))
>
> Yes, our method explicitly enforces the linearity with respect to the latent representations of the prototype pairs. This is achieved by the design of the prediction branch of our model (described in Section 2.2 of our paper), and the minimization of the mean absolute/square error on a training dataset, which ensures that the latent representations of training instances have to vary linearly with respect to the axis defined by their closest prototype pairs in the latent space.
>
> - Why is now a timely moment for this paper? (Question 1)
>
> While it is true that the core ingredients of prototype-based learning have been available for several years, their application has remained largely restricted to classification problems [1–5] or reinforcement learning problems with discrete action spaces [6].
>
> However, a noticeable and important gap remains: prototype-based reasoning has not been meaningfully extended to regression tasks. Unlike classification, where the decision boundaries between classes can be cleanly interpreted via proximity to class prototypes, regression requires a continuous output space and thus demands a fundamentally different formulation of how prototypes are used for prediction. To our knowledge, very few (if any) prior works have addressed this challenge, and those that do have shown weak performance or lacked interpretability (see Table 2 of our paper).
>
> Our work closes this gap by introducing ProtoPairNet, which formulates prototype-based reasoning as comparative rather than absolute—shifting from matching to a single prototype to reasoning based on the relative position between pairs of prototypes. This approach is naturally suited to regression, where inputs can be interpreted as "more like this than that" along a continuous axis. In this sense, now is a timely moment for this work: the community has matured in its understanding of prototype-based interpretability for classification, and the need to extend these benefits to regression is both technically open and practically valuable.
>
> - Do the authors have an explanation for how prototype models performed similarly or even better than the black-box baselines? Would including additional benchmarks help here? (Question 3)
>
> While it is commonly assumed that interpretability comes at the cost of predictive performance, recent research suggests that this tradeoff is not inevitable. As argued in [7], interpretable models can perform on par with black-box models for many real-world tasks. This is because many problems admit a Rashomon set – a large collection of diverse models that achieve similarly high accuracy [8]. Black-box models often find one such solution, but interpretable models can also lie within this set; the difficulty lies not in their existence, but in finding them. Empirical evidence supporting this claim appears in existing prototype-based models [1–6], all of which achieved competitive accuracy while maintaining interpretability.
>
> To further evaluate the generalizability of ProtoPairNet, we have expanded our benchmarking to include additional tasks across diverse domains:
>
> Medical imaging: We applied ProtoPairNet to the APTOS dataset [9] for predicting the severity of diabetic retinopathy from retinal fundus images. This benchmark presents both high-dimensional input (3×224×224) and clinically relevant labels. For the baseline model we used ResNet50.
>
> Tabular regression: We added the California housing prices dataset [10], a classic tabular regression task with continuous outputs and heterogeneous input features. For the black-box baseline, we used a simple fully-connected network with three hidden layers (128, 64, 32 units), each followed by ReLU and 10% dropout. The final layer outputs a single value for predicting house prices.
>
> | Dataset            | Model               | MAE               | R² score         |
> |--------------------|---------------------|-------------------|------------------|
> | **APTOS**          | Baseline (No prototypes) | 0.301 ± 0.008      | 0.825 ± 0.009     |
> |                    | HPN                  | 0.360 ± 0.007      | 0.728 ± 0.006     |
> |                    | ProtoPairNet (ours)        | 0.278 ± 0.028      | 0.837 ± 0.010     |
> | **California housing** | Baseline (No prototypes) | 0.320 ± 0.001      | 0.811 ± 0.004     |
> |                    | HPN                  | 0.275 ± 0.0004     | 0.816 ± 0.001     |
> |                    | ProtoPairNet (ours)        | 0.289 ± 0.003      | 0.796 ± 0.004     |
>
> As shown above, ProtoPairNet performs comparably to baselines in these new benchmarks. We hope these new results address the request for more benchmarks.
>
> - Have the authors tried using any pre-trained backbones? (Question 4)
>
> Yes, our backbone models were pre-trained on ImageNet.
>
> - If the perpendicular drop from the input latent to the prototype axis lies outside the segment formed by the prototype pair, how is this case handled? (Question 5)
>
> Equation (4), which defines prediction from a prototype pair, remains valid even when the perpendicular projection from the input latent to the prototype axis falls outside the prototype segment. As shown in Figures 17 and 18 (Appendix I), ProtoPairNet can predict ages beyond those of the most relevant prototype pair. For example, in Figure 17 (bottom-left), the model predicts age 11.49 for an input of age 12, despite the closest prototype pair having ages 16 and 35. Here, the projected input lies to the left of the prototype segment, which our ProtoPairNet handles elegantly.
>
> [1] Chen, C. et al. (2019). This looks like that: deep learning for interpretable image recognition. NeurIPS.
>
> [2] Nauta, M. et al. (2021). Neural prototype trees for interpretable fine-grained image recognition. CVPR.
>
> [3] Rymarczyk, D. et al. (2022). Interpretable image classification with differentiable prototypes assignment. ECCV.
>
> [4] Ma, C. et al. (2023). This looks like those: Illuminating prototypical concepts using multiple visualizations. NeurIPS.
>
> [5] Ma, C. et al. (2024). Interpretable image classification with adaptive prototype-based vision transformers. NeurIPS.
>
> [6] Ragodos, R. et al. (2022). ProtoX: Explaining a reinforcement learning agent via prototyping. NeurIPS.
>
> [7] Rudin, C. (2019). Stop explaining black box machine learning models for high stakes decisions and use interpretable models instead. Nature Machine Intelligence, 1(5), 206-215.
>
> [8] Breiman, L. (2001). Statistical modeling: The two cultures (with comments and a rejoinder by the author). Statistical Science, 16(3), 199-231.
>
> [9] Kaggle (2019). APTOS 2019 blindness detection. Kaggle.
>
> [10] Pace, R. K., & Barry, R. (1997). Sparse spatial autoregressions. Statistics & Probability Letters, 33(3), 291-297.

---

> > ### Comment · Reviewer_uvMS · 2025-08-09
> > **Thank you for the clarifications.**
> >
> > Thank you for the clarifications. Most of my concerns have now been addressed. I believe that incorporating your responses from the rebuttal into the main paper would strengthen the overall quality and clarity of the work. Therefore, I will retain my original score.

---

> ### Author Response · Authors · 2025-08-05
>
> We hope that our response has helped explain our work's contributions and address your concerns. Please feel free to let us know if you have any further questions. We are very happy to discuss!

---

> > ### Author Response · Authors · 2025-08-07
> > **Follow-up on rebuttal**
> >
> > With the discussion period ending in less than 48 hours, we would like to follow up on our rebuttal and would greatly appreciate it if you could share your thoughts on our rebuttal, so that we may have the opportunity to further clarify and defend our work.
> >
> > Specifically, we would like to initiate a discussion regarding your comment: "If the intent is to increase expressiveness, it would be nice to have an ablation that varies the tuple size from 1 to 2 and to perhaps even higher-order tuples."
> >
> > We would appreciate clarification on what is meant by "tuples" in this context. Our interpretation is that this refers to groups of prototypes -- e.g., using single prototypes (size-1 tuples), pairs (size-2), or higher-order groupings such as triplets.
> >
> > Our ProtoPairNet specifically uses prototype pairs because a single prototype cannot define a direction or axis in the latent space, which is central to our method. Extending to higher-order tuples (e.g., triplets) would introduce additional complexity: in order to define a meaningful direction, the three (or more) prototypes would need to lie on the same line in latent space, which is difficult to enforce and could lead to unstable or uninterpretable behavior. Pairs, in contrast, naturally define an axis and strike a balance between expressiveness and interpretability.
> >
> > For these reasons, we focused our design and experiments on prototype pairs. However, we did include an ablation study for varying number of prototype pairs in our rebuttal.
> >
> > Additional ablation studies of the various loss terms used to train our model can be found in Table 8 in Appendix E of our paper.
> >
> > Please let us know if you have further questions -- we are happy to discuss!

---

> ### Comment · Area_Chair_FUdY · 2025-08-08
>
> Reviewer gJBu: could you please assess the author rebuttal and post your response? please update your rating if your concerns have been resolved. Thank you

---

### Official Review · Reviewer_tFYK · 2025-07-02

**Clarity:** 3
**Significance:** 2
**Originality:** 3
**Rating:** 4
**Confidence:** 5

**Summary:**

This paper introduces ProtoPairNet, a prototype-based framework for interpretable regression. Unlike traditional prototype models that rely on one-to-one comparisons, ProtoPairNet compares inputs to pairs of prototypes, enabling predictions based on the input’s position along the axis between two reference examples. This approach provides directional information needed for accurate continuous predictions. The model also uses a relevance mechanism to select the most appropriate prototype pair for each input, ensuring sparse and interpretable decision-making. Additionally, the authors propose a three-stage training strategy, including a projection of prototypes to real data samples for better visualization. ProtoPairNet is evaluated on age prediction and reinforcement learning-based control tasks, demonstrating competitive performance and improved interpretability over prior methods, as confirmed by multiple user studies.

**Questions:**

1. Table 2: ProtoPariNet seems to significantly outperform other architectures. Are those SOTA architectures, and were they fully fine-tuned for this task?
2. Could you propose more complex benchmark for SOTA comparison? Age prediction seems not sufficient to decide the generalizability of this architecture.
3. MAE and R^2 seem like a limited performance metrics. It would be nice to see a distribution of prediction error, or more exhaustive metrics that describe it (along with comparison to other models).
4. Would be nice to see some attribution method (like LRP or Integrated Gradients) to justify the explanation reasoning (Is the model really focusing on age, or maybe something else?)

**Ethical Concerns:**

["NO or VERY MINOR ethics concerns only"]

**Final Justification:**

The paper presents an interesting and to some degree interpretable approach to regression using prototype pairs, offering a fresh perspective on explainable AI for continuous prediction tasks. The authors strengthened the submission by additional evaluations. However, limitations remain in the reliance on global prototypes without built-in fine-grained attribution and the mixed performance across domains, justifying a borderline accept.

**Limitations:**

The authors have noted the limitation concerning global prototypes rather than prototype-part based ones (or attribution-based). However, my concern lies with the benchmarking, specifically the datasets and evaluation metrics employed, as well as the generalizability of the approach to more complex scenarios.

**Paper Formatting Concerns:**

None.

**Quality:**

2

**Strengths And Weaknesses:**

Strengths:
- Interesting approach for prototypical regression: Prototype-based models have been widely used for classification, but very few extend to regression. ProtoPairNet addresses this by introducing a structured, interpretable framework for continuous prediction tasks. This makes it a novel and valuable contribution to the field of explainable AI.
- Interpretable Prototypes: Prototypes in ProtoPairNet are projected onto actual training instances, making them easy to visualize and understand. Unlike abstract latent vectors, these prototypes can be shown as real images. This adds transparency and helps users intuitively understand how the model arrived at its prediction.
- Backbone compatibility: ProtoPairNet is compatible with various standard architectures like ResNet, EfficientNet, and Vision Transformers. This flexibility makes it easy to apply to various domains without redesigning the core model from scratch. This modular design also allows for future extensions.
- Selection mechanism for explanation clarity: ProtoPairNet uses a selection mechanism to select the most relevant prototype pair for each prediction. This reduces noisy explanations and ensures that the reasoning remains focused and interpretable.

Weaknesses
- Limited benchmarking: ProtoPairNet is only tested on two tasks: age prediction and a car racing control task. These are relatively constrained problems that may not reflect the challenges of more complex, noisy, or high-dimensional regression domains. Without broader evaluation, the model’s generalizability remains uncertain.
- Global explanations: Unlike part-prototype models that highlight specific input regions (e.g. facial features), ProtoPairNet uses global prototypes (entire images). This means it cannot indicate which parts of an input were most important for the prediction. As a result, it lacks fine-grained, spatial explanations, which limits its usefulness and interpretability, as we cannot be sure which features are contributing to the prototypical reasoning. It also makes the interpretation subjective to the user, which might be misaligned with how the model really works.
- Limited use of multiple pairs per prediction: The model predicts using only one prototype pair per input. While this improves interpretability, it may not capture all relevant aspects of the input when multiple semantic factors are relevant. This reasoning could reduce prediction quality in more complex scenarios.

---

> ### Author Rebuttal · Authors · 2025-07-31
>
> - Limited benchmarking and more complex benchmark (Weakness 1, Question 2, Limitations):
>
> Thank you for highlighting this important point. While our initial experiments focused on age prediction and car racing, it is worth noting that both tasks involve high-dimensional visual inputs – 3×128×128 images for age prediction and 4×96×96 stacked frames for CarRacing – making them non-trivial and representative of real-world perceptual challenges. To further evaluate the generalizability of ProtoPairNet, we have expanded our benchmarking to include two additional tasks across diverse domains:
>
> Medical imaging: We applied ProtoPairNet to the APTOS dataset [1] for predicting the severity of diabetic retinopathy from retinal fundus images. This benchmark presents both high-dimensional input (3×224×224) and clinically relevant labels. For the baseline model we used ResNet50.
>
> Tabular regression: We applied ProtoPairNet to the California housing dataset [2], a classic tabular regression task with 8-dimensional inputs and continuous outputs. The baseline is a fully connected network with three hidden layers (128, 64, 32 units), ReLU activations, 10% dropout, and a single output neuron for predicting house prices.
>
>
> | Dataset            | Model               | MAE               | R² score         |
> |--------------------|---------------------|-------------------|------------------|
> | **APTOS**          | Baseline (No prototypes) | 0.301 ± 0.008      | 0.825 ± 0.009     |
> |                    | HPN                  | 0.360 ± 0.007      | 0.728 ± 0.006     |
> |                    | ProtoPairNet (ours)        | 0.278 ± 0.028      | 0.837 ± 0.010     |
> | **California housing** | Baseline (No prototypes) | 0.320 ± 0.001      | 0.811 ± 0.004     |
> |                    | HPN                  | 0.275 ± 0.0004     | 0.816 ± 0.001     |
> |                    | ProtoPairNet (ours)        | 0.289 ± 0.003      | 0.796 ± 0.004     |
>
> As shown above, ProtoPairNet performs comparably to baselines in these new benchmarks. We hope these new results address the request for more benchmarks.
>
> - Global explanations and attribution method (Weakness 2, Question 4):
>
> Thank you for the thoughtful comment. Prior work on prototype-based deep neural networks indeed spans two major approaches: using prototypes defined at the global (entire image) level [3, 4] and those defined at the local (patch or part) level [5]. Our choice to adopt global prototypes aligns with the former line of work and is a deliberate design decision, primarily due to the nature of our data and labels. Specifically, for age predictions, learning semantically meaningful part prototypes can be challenging, since a 50-year-old may have facial features that resemble someone younger, making part-based prototypes less reliable.
>
> That said, we fully agree with the concern about the lack of fine-grained spatial attribution. While ProtoPairNet’s global prototypes naturally provide semantic interpretability at the image level (e.g., "this face resembles an older person"), they do not by themselves reveal which parts of the input contributed most to the prediction. Fortunately, this limitation can be addressed via post-hoc attribution methods. For instance, occlusion-based sensitivity analysis [6] or region perturbation methods [7] can help localize which input regions most influence the similarity (or dissimilarity) to a given prototype pair. This approach has precedent: Ragodos et al. [4] also use global prototypes (entire game frames) but apply post-hoc attribution to identify salient regions that drive prototype similarity.
>
> In our setting, to identify which input region a prototype pair attends to for a given input image, we sequentially masked input patches and measured the change in its prediction as an importance score. This patch occlusion method, though not built into the core architecture, provides spatial explanations without requiring unreliable part-level supervision. We analyzed the top-10 most important patches across 5 randomly chosen test images and found that high-impact regions consistently fall within rows 2–6 and columns 2–6 of the 8×8 patch grid—corresponding to key facial landmarks such as the eyes, nose, lips, and jawline. This suggests the model consistently focuses on semantically meaningful features.
>
> Quantitatively, as shown in the table below, the average change in prediction across occluded patches ranged from 0.38 to 1.07, with maximum changes per image between 1.82 and 4.40, confirming that these regions exert a significantly greater influence on the model’s output. In all cases, these regions corresponded to one of the facial landmarks.
>
> | Test image   | Average prediction change | Max prediction change |
> |---------------|----------------|-------------|
> | Test image 1  | 0.525          | 2.737       |
> | Test image 2  | 0.773          | 4.153       |
> | Test image 3  | 0.378          | 1.820       |
> | Test image 4  | 1.071          | 4.400       |
> | Test image 5  | 0.884          | 3.438       |
>
> - Limited use of multiple pairs per prediction (Weakness 3):
>
> Thank you for the insightful comment. We agree that relying on a single prototype pair may limit expressiveness when multiple semantic factors are relevant. However, this design choice enhances interpretability by encouraging reasoning along a single, identifiable axis.
>
> To evaluate the trade-off, we conducted an ablation study in which we replaced the Gumbel-softmax-based prototype pair relevance branch with (1) a standard softmax and (2) a simple average of all prototype pairs’ predictions. For a given input, the standard softmax weights each prototype pair’s prediction proportional to the negative exponent of the distance between the input’s latent representation to the pair’s axis, while a simple average simply outputs the arithmetic average of all prototype pairs’ predictions. In both cases, the predicted value uses more than one prototype pair. We found that using multiple pairs did not significantly improve performance across our benchmarks, suggesting that a single pair often captures the dominant factor driving the prediction (see the table below).
>
> This observation is also consistent with findings from prior work [8], which showed that projecting to a single prototype, rather than a weighted combination of multiple prototypes, has minimal impact on performance (see Table 9 of that paper).
>
> Hence, we view the use of the most relevant pair not necessarily as a disadvantage, since it maintains predictive performance and at the same time improves interpretability.
>
> | ProtoPairNet Predictions | MAE             | R² score         |
> |--------------------------|----------------|------------------|
> | Standard Softmax         | 4.49 ± 0.007    | 0.877 ± 0.0006   |
> | Simple Average           | 4.48 ± 0.003    | 0.878 ± 0.0004   |
> | Gumbel Softmax (ours)| 4.52 ± 0.010    | 0.876 ± 0.0009   |
>
> - Comparison with other architectures in Table 2 (Question 1):
>
> Yes, the baseline architectures we compared against are state-of-the-art visual regression models, and were fully fine-tuned for each regression task using standard training protocols and hyperparameter sweeps to ensure fair comparison.
>
> It is important to note that, unlike classification, there are relatively few prototype-based architectures specifically designed for visual regression tasks. Our ProtoPairNet is designed specifically for prototype-based regression, and its performance gains reflect the strength of using prototype-pair reasoning in these settings.
>
> - Performance evaluation metrics (Question 3, Limitations):
>
> Thank you for the suggestion. For the age prediction task, we did an analysis of the distribution of mean average errors over different age groups using the ResNet50-based ProtoPairNet and the standard (black-box) ResNet50 models. The results are summarized in a table.
>
> | Age group   | ProtoPairNet MAE (ours)   | ResNet50 MAE        | HPN MAE            | Count |
> |-------------|----------------------|----------------------|---------------------|--------|
> | [0, 15)     | 2.06 ± 0.14  | 1.93 ± 0.03  | 2.57 ± 0.44  | 366    |
> | [15, 30)    | 3.52 ± 0.17  | 3.48 ± 0.16  | 4.36 ± 0.37   | 830    |
> | [30, 45)    | 4.80 ± 0.15  | 5.46 ± 0.05 | 5.67 ± 0.54   | 570    |
> | [45, 60)    | 7.14 ± 0.33  | 7.60 ± 0.04 | 7.43 ± 0.38  | 335    |
> | [60, 75)    | 7.37 ± 0.77 | 6.83 ± 0.18  | 7.52 ± 0.06   | 169    |
> | [75, 90)    | 10.26 ± 1.32  | 10.18 ± 0.43  | 10.80 ± 1.10 | 57     |
>
> As shown in the above table, the ResNet50-based ProtoPairNet performs better than the black-box ResNet50 for two age groups [30, 45) and [45, 60), and performs similarly for the other age groups.
>
> [1] Kaggle (2019). APTOS 2019 blindness detection. Kaggle.
>
> [2] Pace, R. K., & Barry, R. (1997). Sparse spatial autoregressions. Statistics & Probability Letters, 33(3), 291-297.
>
> [3] Kenny, E. M., Tucker, M., & Shah, J. (2023). Towards interpretable deep reinforcement learning with human-friendly prototypes. International Conference on Learning Representations.
>
> [4] Ragodos, R., Wang, T., Lin, Q., & Zhou, X. (2022). ProtoX: Explaining a reinforcement learning agent via prototyping. Advances in Neural Information Processing Systems, 35, 27239-27252.
>
> [5] Chen, C., Li, O., Tao, D., Barnett, A., Rudin, C., & Su, J. K. (2019). This looks like that: deep learning for interpretable image recognition. Advances in neural information processing systems, 32.
>
> [6] Zeiler, M. D., & Fergus, R. (2014). Visualizing and understanding convolutional networks. European Conference on Computer Vision.
>
> [7] Petsiuk, V., Das, A., & Saenko, K. (2018) RISE: Randomized Input Sampling for Explanation of Black-box Models. British Machine Vision Conference.
>
> [8] Hong, D., Wang, T., & Baek, S. (2023). Protorynet-interpretable text classification via prototype trajectories. Journal of Machine Learning Research, 24(264), 1-39.

---

> > ### Comment · Reviewer_tFYK · 2025-08-05
> >
> > You strengthened the submission with additional evaluations, which is why I increased my score to borderline acceptable. However, limitations remain due to reliance on global prototypes without built-in fine-grained attribution, as well as mixed performance across domains.

---

> ### Author Response · Authors · 2025-08-07
> **Thank you for your feedback**
>
> Thank you for your thoughtful feedback and for recognizing the additional evaluations we provided. We appreciate your willingness to increase our paper's score.

---

### Official Review · Reviewer_uvMS · 2025-07-05

**Clarity:** 2
**Significance:** 3
**Originality:** 3
**Rating:** 4
**Confidence:** 2

**Summary:**

This paper introduces ProtoPairNet, a novel interpretable architecture that combines deep learning with case-based reasoning to predict continuous targets, extending prototype-based models from solving classification tasks with discrete outputs to continuous targets such as regression tasks.
Specifically, ProtoPairNet leverages prototypical pairs as ProtoPair axis in the latent space: given an input, it makes predictions based on the relative positioning along the ProtoPair axis defined by its most relevant prototype pair. The architecture comprises 1) a prediction branch that uses geometric projection onto axis between prototypical pairs, and 2) a pair relevance branch that selects the most relevant axis for a given sample via Gumbel-softmax.
ProtoPairNet is evaluated on two domains: age prediction (a supervised learning task) and car racing (a behavioral cloning task with a reinforcement learning expert), achieving performance competitive with black-box baselines, while providing transparent, case-based reasoning for predictions.

**Questions:**

1.	Generalization Beyond Vision: Have the authors considered testing ProtoPairNet on non-visual regression tasks (e.g., tabular, timeseries)? If so, what difficulties arise, and how can pair-axes be made interpretable in domains without natural prototypes?
2.	Out-of-Distribution Reasoning: How does the model handle inputs whose latent representation falls far outside any prototype pair axis? Is there any mechanism for cases when the prototype pairs are insufficient to cover the data manifold?
3.	Potential Failure Modes: This paper would benefit from a more thorough discussion and analysis of potential failure modes.
4.	Baseline Clarification: The description of baseline in Table 1 is not sufficiently clear.

**Ethical Concerns:**

["NO or VERY MINOR ethics concerns only"]

**Final Justification:**

Most of my concerns have now been addressed. I believe that incorporating your responses from the rebuttal into the main paper would strengthen the overall quality and clarity of the work. Therefore, I will retain my original score.

**Limitations:**

yes

**Quality:**

2

**Strengths And Weaknesses:**

Strengths:
- Quality
1. The paper provides very concrete and detailed mathematical derivations as proof, demonstrating strong theoretical support for the proposed method.
2. The paper provides very concrete and detailed mathematical derivations as proof, demonstrating strong theoretical support for the proposed method.
3. The approach is rigorous, including a three-stage training process and a non-trivial prototype projection algorithm to ensure semantic and geometrical coherence.
-Clarity
1. The paper is well written, with clear and thorough explanations.
2. All figures and tables contribute positively to readers’ understanding.
- Significance
1. The paper tackles an important problem, namely extending prototype-based models from discrete classification to interpretable regression tasks using reasoning-based approaches.
- Originality
1. The model architecture and training paradigm can be considered novel.

Weaknesses
- Quality
1. The experimental results, while competitive, are only evaluated on a limited set of regression tasks (face age, car racing in paper). Vision is heavily emphasized, no "classic" tabular regression or high-dimensional time-series regression tasks are included.
2. There is a lack of discussion of potential failure modes or cases where one-to-one or traditional regression models might outperform (e.g., highly nonlinear target relationships, out-of-distribution scenarios).

- Clarity
1. Some descriptions, such as the prototype projection (Section 2.4), are mathematically dense and could be made more accessible, 2. The description of baselines in Table 1 is not sufficiently clear.
- Significance
1. While user studies are carefully executed, all evaluation hinges on just two datasets. It is not fully clear how well this method generalizes to hard regression tasks (e.g., non-vision or highly nonlinear problems).
2. ProtoPairNet’s interpretability hinges on whether its selected prototype pairs are semantically coherent and representative. If the nearest prototype pair for an input does not correspond to a meaningful axis or concept in the data space, the resulting explanation may lack clarity and fail to provide useful human insight.
- Originality
1. Underlying notions are well established, though the extension to pairs and projection-based reasoning is new.

---

> ### Author Rebuttal · Authors · 2025-07-31
>
> - Limited benchmarking and generalization beyond vision (Quality weakness 1, Significance weakness 1, Question 1):
>
> Thank you for the suggestion! To further evaluate the generalizability of ProtoPairNet, we have expanded our benchmarking to include two additional tasks across diverse domains:
>
> Tabular regression: Following your suggestion, we added the California housing prices dataset [1], a classic tabular regression task with continuous outputs and heterogeneous input features. For the black-box baseline, we used a simple fully-connected network with three hidden layers (128, 64, 32 units), each followed by ReLU and 10% dropout. The final layer outputs a single value for predicting house prices.
>
> Medical imaging: We applied ProtoPairNet to the APTOS dataset [2] for predicting the severity of diabetic retinopathy from retinal fundus images. This benchmark presents both high-dimensional input (3×224×224) and clinically relevant labels. For the baseline model we used ResNet50.
>
> | Dataset            | Model               | MAE               | R² score         |
> |--------------------|---------------------|-------------------|------------------|
> | **California housing** | Baseline (No prototypes) | 0.320 ± 0.001      | 0.811 ± 0.004     |
> |                    | HPN                  | 0.275 ± 0.0004     | 0.816 ± 0.001     |
> |                    | ProtoPairNet (ours)        | 0.289 ± 0.003      | 0.796 ± 0.004     |
> | **APTOS**          | Baseline (No prototypes) | 0.301 ± 0.008      | 0.825 ± 0.009     |
> |                    | HPN                  | 0.360 ± 0.007      | 0.728 ± 0.006     |
> |                    | ProtoPairNet (ours)        | 0.278 ± 0.028      | 0.837 ± 0.010     |
>
> In a tabular domain such as the California housing dataset, we can interpret prototype pairs through their feature values and treat the pair-axes as a semantic contrast between representative conditions (e.g., high vs. low income regions in tabular data). While direct visualization is not possible, these axes can still be interpretable through feature-level analysis or domain-specific context.
>
> While ProtoPairNet was primarily designed for vision-based regression tasks, our additional benchmark on the tabular California housing data demonstrates that our ProtoPairNet is more flexible and generalizable than originally intended.
>
> As shown in the table above, ProtoPairNet performs comparably to baselines in the two additional benchmarks. We hope the new experiments satisfy the requirement of broader evaluation and generalization beyond vision.
>
> - Out-of-distribution reasoning (Question 2):
>
> Inputs whose latent representations fall far outside any ProtoPair axis indeed raise important considerations with regard to out-of-distribution (OOD) reasoning, but the challenge of OOD samples is not unique to our model – it is a fundamental limitation faced by virtually all machine learning systems. No model can be expected to reliably reason about regions of the data space that are sparsely populated or unsupported by training examples. In this sense, the concern applies broadly and is not specific to our approach.
>
> However, ProtoPairNet is particularly well-positioned to address this issue. Because its predictions are based on distances to prototype pairs, the model provides a direct and interpretable signal for when an input lies far from any axis of comparison. This allows for a natural mechanism for out-of-distribution (OOD) detection: by measuring the distance from a test point to its nearest ProtoPair axis and flagging cases where this distance exceeds a threshold, we can identify inputs that fall outside the region where the model is expected to reason reliably. While we have not yet implemented such an OOD detection module, the structure of ProtoPairNet makes it amenable to this kind of extension, offering a promising path toward increased robustness and interpretability.
>
> - Potential failure modes (Quality weakness 2, Question 3):
>
> Thank you for the suggestion. We would include a more thorough discussion of potential failure modes in our final paper. Here are some potential failure modes of our ProtoPairNet model:
>
> 1. Out-of-distribution inputs: As previously discussed, if an input's latent representation falls far from all prototype pair axes, the model may produce unreliable outputs. Without explicit mechanisms to detect or adapt to such inputs, the reasoning based on prototype pairs may fail. However, we can use a distance-based threshold to flag such cases and consider this an important direction for future development.
>
> 2. Insufficient prototype coverage: If the learned prototype pairs fail to span the full diversity of the data manifold, the model may generalize poorly to underrepresented regions. Learning a sufficient number of diverse prototype pairs should address this issue. In our experiments, we determined the ideal number of prototype pairs for each task using a validation set.
>
> - Clarity (Clarity weakness 1, Question 4):
>
> We will describe prototype projection (Section 2.4) and the baseline models in Table 1 more clearly in our final paper.
>
> - Semantic coherence of prototype pairs (Significance weakness 2):
>
> We agree that the quality of explanations depends on whether the selected prototype pairs form meaningful, interpretable axes. To encourage this, ProtoPairNet is explicitly designed to learn semantically useful latent axes through training, ground each prototype pair in real training examples, and select the most relevant pair at inference time using Gumbel-softmax, which focuses reasoning on a single interpretable dimension. To validate the semantic coherence of prototype pairs, we conducted a user study (Study 1) where participants were asked to choose which of two prototype pairs was more helpful for the age prediction task. The results show that users largely agree with the model’s prototype-pair selection, suggesting that the axes learned by ProtoPairNet often align with human-understandable concepts.
>
> - Originality (Originality weakness 1):
>
> We agree that ProtoPairNet builds on well-established ideas in prototype-based learning and interpretable modeling. However, most prior work in this space has focused almost exclusively on classification, where similarity to a single prototype can directly inform discrete label decisions. In contrast, regression remains relatively underexplored in prototype-based learning, despite its widespread relevance. Our main contribution is to extend prototype reasoning to regression by introducing prototype pairs and projection-based reasoning. This allows our ProtoPairNet to reason about where an input lies along a semantic axis defined by two prototypes with continuous labels. It addresses key challenges that arise in regression, such as ambiguity in single-prototype similarity and interference from unrelated prototypes, which are not effectively handled by existing prototype-based classification methods.
>
> [1] Pace, R. K., & Barry, R. (1997). Sparse spatial autoregressions. Statistics & Probability Letters, 33(3), 291-297.
>
> [2] Kaggle (2019). APTOS 2019 blindness detection. Kaggle.

---

> > ### Author Response · Authors · 2025-08-05
> >
> > We hope that our response has helped explain our work's contributions and address your concerns. Please feel free to let us know if you have any further questions. We are very happy to discuss!

---

> > ### Comment · Reviewer_uvMS · 2025-08-06
> > **Thank you for the clarifications**
> >
> > Thank you for the clarifications. My concerns have now been addressed. I believe that incorporating your responses from the rebuttal into the main paper would strengthen the overall quality and clarity of the work. Therefore, I will retain my original score.

---

> ### Author Response · Authors · 2025-08-07
> **Thank you for your feedback**
>
> Thank you for your thoughtful feedback and for taking the time to review our rebuttal. We are glad to hear that your concerns have been addressed. We will incorporate the responses from our rebuttal into the main paper to improve its quality and clarity.

---

### Note · Authors · 2025-08-15

Dear reviewers and ACs,

Thank you for your thoughtful reviews and discussions. To support the final evaluation, we summarize our contributions and key rebuttal points below.

**Contributions**:

- Novel architecture for continuous prediction: We propose ProtoPairNet, a novel and interpretable model that extends prototype-based reasoning to continuous prediction by using prototypical pairs. This enables directional reasoning required for regression, which prior prototype models lack.

- Intuitive design with mathematical grounding: ProtoPairNet is based on a simple geometric intuition -- comparing an input's position relative to prototype pairs in a learned latent space, and is formalized with clear mathematical derivations for transparency and rigor.

- Strong performance with enhanced interpretability: ProtoPairNet performs on par with black-box models and offers human-interpretable reasoning, as confirmed by both quantitative results and extensive user studies.

**Key rebuttal points**:

- Additional benchmarking: We added two new tasks -- APTOS (medical imaging) and California housing (tabular regression), and found that ProtoPairNet performs comparably to baselines. For age prediction, we analyzed MAE across age groups, where ProtoPairNet outperforms the black-box model in two age groups and performs similarly in others.

- Part attribution: To identify which input region contributed most to a prediction, we conducted occlusion sensitivity analysis on 5 test images, showing that ProtoPairNet attends to semantically meaningful regions.

- Ablations: We replaced Gumbel-softmax pair relevance with (1) a standard softmax and (2) a simple average. The two variants performed similarly as ProtoPairNet, suggesting that using a single prototype pair per prediction (as in ProtoPairNet) is sufficient. See Appendix E for more ablations.

- Timeliness of our paper: While prototype-based learning is well-studied in classification, it has rarely been extended to regression. ProtoPairNet addresses this gap with a novel formulation tailored to continuous outputs, making it a timely contribution to the field.

- Other clarifications: We addressed questions on OOD reasoning, prototype pair learning, the linearity assumption of the latent space, inputs outside prototype pair segments, prototype diversity, user studies, and computational efficiency.

We believe that our rebuttal has adequately addressed the reviewers’ concerns. Thank you for your time and consideration.

---

### Decision · Program_Chairs · 2025-09-17

**Decision:**

Accept (poster)

**Comment:**

This paper introduces ProtoPairNet, a novel architecture that extends interpretable, prototype-based reasoning to regression and continuous control tasks. The core idea is to move beyond the traditional one-to-one comparisons used in prototype-based classification and instead use "prototypical pairs". These pairs define axes in the latent space, and the model makes predictions based on an input's geometric projection onto the most relevant axis. This is an interesting solution to a key challenge in interpretable regression: capturing the directional information needed for continuous predictions. The method is validated on multiple tasks including age prediction and a car racing behavioral cloning task, where it achieves performance competitive with black-box models while providing transparent, case-based explanations.

The primary strength of this paper is its novel and elegant conceptual contribution. As noted by multiple reviewers (tFYK, gJBu), extending prototype-based methods to regression is an important and underexplored area. ProtoPairNet's use of paired prototypes is a simple yet powerful idea that is well-grounded mathematically and provides an intuitive geometric interpretation for its reasoning process. Crucially, the paper demonstrates that this interpretability does not come at a significant cost to performance; the model achieves results on par with, and in some cases exceeding, its black-box counterparts. The interpretability claims are further supported by extensive user studies, which confirm that the model's reasoning aligns well with human intuition and is preferred over single-prototype explanations.

The main weakness, shared by several reviewers (uVMS, tFYK), is the limited scope of the experimental validation. The evaluation is primarily focused on two domains, and a broader set of benchmarks, including non-visual tasks, would strengthen the claims of generality. Another valid point, raised by reviewer tFYK, is that the interpretability is based on global prototypes (entire images) and lacks the fine-grained, part-based attribution of some other prototype models. However, the authors made a great effort to address these concerns during the rebuttal. They added new experiments on tabular and medical imaging datasets, showing competitive performance and broadening the evaluation scope. They also conducted a post-hoc occlusion analysis to provide more evidence that the model focuses on semantically relevant regions.

Overall, this is a strong paper that introduces a novel, well-motivated, and thoroughly evaluated method for a challenging problem. While the evaluation could be broader, the core contribution is significant enough to warrant acceptance. The authors' proactive and comprehensive rebuttal further strengthened the submission by addressing the main reviewer concerns.